# Parechovirus infection in human brain organoids: host innate inflammatory response and not neuro-infectivity correlates to neurologic disease

Pamela E. Capendale[1,2,7], Inés García-Rodríguez [1,2,7], Anoop T. Ambikan[3], Lance A. Mulder[1,2], Josse A. Depla[2,4], Eline Freeze[1,2], Gerrit Koen[2], Carlemi Calitz [1,2], Vikas Sood[3,5], Renata Vieira de Sá[2,3], Ujjwal Neogi [3], Dasja Pajkrt [1,2], Adithya Sridhar [1,2,6,7] & Katja C. Wolthers [2,7] ✉

Picornaviruses are a leading cause of central nervous system (CNS) infections. While genotypes such as parechovirus A3 (PeV-A3) and echovirus 11 (E11) can elicit severe neurological disease, the highly prevalent PeV-A1 is not associated with CNS disease. Here, we expand our current understanding of these differences in PeV-A CNS disease using human brain organoids and clinical isolates of the two PeV-A genotypes. Our data indicate that PeV-A1 and A3 specific differences in neurological disease are not due to infectivity of CNS cells as both viruses productively infect brain organoids with a similar cell tropism. Proteomic analysis shows that PeV-A infection significantly alters the host cell metabolism. The inflammatory response following PeV-A3 (and E11 infection) is significantly more potent than that upon PeV-A1 infection. Collectively, our findings align with clinical observations and suggest a role for neuroinflammation, rather than viral replication, in PeV-A3 (and E11) infection.

Parechovirus A (PeV-A), formerly known as human parechoviruses, are common childhood pathogens in the *Picornaviridae* family with a potential for severe clinical manifestations, mainly in infants[1]. PeV-A has been observed to circulate in several countries, such as the Netherlands[2], Japan[3], and the United States of America (USA)[4]. PeV-As are closely related to the *Enterovirus* (EV) genus, showing similar clinical characteristics and outbreak potential. PeV-As are as prevalent as EVs and are the second leading cause of viral CNS infections in neonates[1]. The striking parallelism between PeV-A and EVs is highlighted by the initial classification of PeV-A as echoviruses[5], a polyphyletic group of "orphan" viruses within the EV genus that includes neurotropic viruses such as echovirus 11 (E11)[6–8].

The PeV-A species is subdivided into 19 genotypes, with the most prevalent genotypes being PeV-A1 and PeV-A3[9]. Although both PeV-A1 and PeV-A3 can elicit gastrointestinal and respiratory disease, only PeV-A3 is predominantly associated with central nervous system (CNS) disease[10,11]. Several PeV-A3 outbreaks have been reported globally, with the most recent one in the USA in 2022[12]. After infection of the primary

[1]OrganoVIR Labs, Emma Children's Hospital, Department of Pediatric Infectious Diseases, Amsterdam UMC, Academic Medical Center, Amsterdam Institute for Infection and Immunity, Amsterdam Institute for Reproduction and Development, University of Amsterdam, Meibergdreef 9, Amsterdam, The Netherlands. [2]OrganoVIR Labs, Department of Medical Microbiology, Amsterdam UMC, Academic Medical Center, Amsterdam Institute for Infection and Immunity, University of Amsterdam, Meibergdreef 9, Amsterdam, The Netherlands. [3]The Systems Virology Lab, Division of Clinical Microbiology, Department of Laboratory Medicine, Karolinska Institutet, ANA Futura, Campus Flemingsberg, Stockholm, Sweden. [4]UniQure Biopharma B.V., Department of Research & Development, Paasheuvelweg 25A, Amsterdam, The Netherlands. [5]Department of Biochemistry, School of Chemical and Life Sciences, Jamia Hamdard, New Delhi, India. [6]Emma Center for Personalized Medicine, Amsterdam UMC, Amsterdam, The Netherlands. [7]These authors contributed equally: Pamela E. Capendale, Inés García-Rodríguez, Adithya Sridhar, Katja C. Wolthers. ✉e-mail: k.c.wolthers@amsterdamumc.nl

replication sites (airway and/or intestinal epithelium), the virus can reach the bloodstream causing sepsis-like illnesses and infect other organs[10], causing CNS-related diseases like transient paralysis[13], encephalitis[1,14], and meningitis[10,14]. Most of these cases occur in infants younger than three months of age[10,15]. In addition to these acute clinical manifestations, long-term neurological sequelae are reported, such as neurodevelopmental delays, impairment in auditory functions, or gross motor function delay[16,17]. Despite the remarkable differences in short- and long-term morbidity between PeV-A1 and A3, the underlying reasons for the differences are yet to be discovered.

A possible explanation for this genotype-specific difference in disease could be related to a preference for PeV-A3 to infect (other) CNS cell types compared to PeV-A1. We previously reported that PeV-A3 strains showed higher replication kinetics in a neural cell line (SH-SY-5Y) compared to PeV-A1 strains[18]. Another potential explanation is structural differences in the receptor-binding region. The VP1 of PeV-A1, but not of PeV-A3, contains an Arginyl-glycyl-aspartic acid (RGD) motif, which enables PeV-A1 to bind to cell membrane-bound integrins[19,20] suggesting differential receptor usage of PeV-A1 and -A3 for entry. This differential use could lead to a difference in cell tropism and subsequent disease. However, a host membrane protein, myeloid-associated differentiation marker (MYADM), was recently shown as the entry receptor for both genotypes A1 and A3[21]. Lastly, we reported a higher inflammatory response due to PeV-A3 infection in primary human airway epithelial (HAE) cultures[22] as compared to PeVA1 infection. This difference in genotype-specific inflammatory responses may also explain the differences in PeV-A1 and PeV-A3-induced CNS disease.

Studies on PeV-A neurologic disease have been limited to immortalized cell lines, and one study was performed using an animal model[23]. Studying PeV-A pathogenesis on relevant human models of the CNS, such as brain organoids, can provide insights in the natural human neurodevelopmental context. Organoids are 3D cell culture models that recapitulate the cellular composition, structure, and complexity of the organ they mimic. In the case of brain organoids, they are valuable tools in modeling human neurodevelopment[24]. There are several types of organoids that mimic the brain with the two main subtypes being unguided neural organoids (UNOs)[25] and regionalized neural organoids. UNOs are three-dimensional (3D) structures generated from human pluripotent stem cells (hPSCs) that recapitulate characteristics of the developing human brain[26]. Compared to regionalized organoids, UNOs encompass broader characteristics of the human CNS with different cell types and regions present as seen during human neurodevelopment[27]. UNOs mature in stages and show genetic features similar to the developing human embryonic brain[26,28], including different neuronal and glial cell types, but usually lacking immune cells[29].

UNOs and other brain organoid models have previously been used to study infection of various viruses[30]. For example, UNOs recapitulate the Zika virus (ZIKV)-induced fetal microcephaly observed in patients[31,32]. Similarly, UNOs have allowed for the study of herpes simplex virus 1 (HSV-1) reactivation[33], and when infected with human cytomegalovirus (HCMV), UNOs showed similar patterns to clinical brain specimens[34]. Multiple advantages of organoids over conventional models have been demonstrated for studying CNS-related viruses[30]. They have proven to be of great value in recapitulating cellular tropism and the effect of infection on cellular organization[35–38]. These benefits show great promise for addressing the questions regarding the PeV-specific CNS disease.

In this study, we use UNOs to study the effects of viral infection with genotypes PeV-A1 and PeV-A3. Infection of UNOs with E11 is included as a control of a neuropathogenic virus that causes clinically similar neurological disease as described for PeV-A3[39]. Our aim is to identify differences in neuropathological effects caused by neuropathogenic (par)echoviruses compared to the non-neuropathogenic

PeV-A1 genotype. We aim to elucidate the mechanisms behind this by evaluating the viral replication kinetics, cell tropism, and host (inflammatory) response.

## Results

### Lab-adapted strains of PeV-A1, PeV-A3, and E11 infect and replicate in UNOs

UNOs were cultured for 67 days (Supplementary Fig. 1a). To determine the cytoarchitecture and the cell types present in the UNOs, they were assessed by immunofluorescence. At this UNO developmental age, we expected the presence of progenitor zones surrounded by self-organized patterns of neurons and astrocytes[26]. Indeed, the generated UNOs featured typical ventricular-like zones (VZs) with neural progenitor cells (NPCs) (SOX2+) in the center. These VZs were surrounded by radially distributed mature neurons (MAP2+) and specific astrocyte-rich regions (GFAP+) (Supplementary Fig. 1b and Supplementary Movie 1). Moreover, the cortical regions within the UNOs displayed a stereotypical layered organization of the developing human brain as previously described[40,41]. These cortical regions included cells positive for neural progenitor cell marker PAX6+ in the center (VZ). We observed organized cell layers surrounding the VZ that expressed neural markers specific for early-born deep-layer neurons (CTIP2+), and late-born superficial layer neurons (SATB2+)[26,40] (Supplementary Fig. 1c). This indicates proper development and layer organization that is observed in the developing human fetal brain[41], hence presenting a good model for the study of viral CNS infection in neonates.

To compare infection dynamics of the two PeV-A genotypes, the 67-day-old UNOs were inoculated with lab-adapted strains of PeV-A1 (strain Harris), PeV-A3 (strain 152037), and E11 (strain 50473) (Fig. 1a). We observed significant replication of E11 in UNOs with peak copy numbers on day 3. For both PeV-A1 and PeV-A3 infected UNOs, a significant increase in viral RNA copies was seen over time. However, the kinetics of PeV-A1 and PeV-A3 replication were different. PeV-A1 showed similar replication kinetics to that of E11, while PeV-A3 showed a slower and lower replication than PeV-A1 and E11 (Fig. 1b). The increase in RNA copies was associated with active viral replication, as we observed a reduction in viral RNA copies over time when the viruses were heat-inactivated before inoculation (Supplementary Fig. 2). Finally, a 50% tissue culture infectious dose (TCID50) assay was performed to confirm the generation of infectious viral particles. In accordance with the RT-qPCR data, there was a significant increase in TCID50 over time for all three viruses, indicative of the presence of infectious viral particles (Fig. 1c).

### No difference in cell tropism for lab-adapted strains of PeV-A1, PeV-A3, and E11

Immunocytochemistry was used to visualize the viral tropism in UNOs and to identify possible changes in the organoid architecture often accompanying viral infection in different brain organoid models[34,42,43]. We did not observe any major changes in the cytoarchitecture of the UNOs due to the viral infection (Supplementary Figs. 3 and 4). UNOs infected with PeV-A1 Harris (Supplementary Fig. 3b) or PeV-A3 152037 (Supplementary Fig. 3c) showed positive dsRNA (indicative of viral infection) in astrocyte (GFAP+) and neuron (MAP2+) rich areas. Infection of GFAP and MAP2-positive cells was confirmed by the use of orthogonal sections (Fig. 2 and Supplementary Fig. 5). Moreover, we did not observe dsRNA within VZs, suggesting that NPCs are not susceptible to PeV-A1 and A3 infection (Supplementary Fig. 3). Similarly, E11 50473 was mainly found in GFAP+ and MAP2+ areas (Supplementary Fig. 4). We further confirmed this with virus-specific antibodies (validated in Supplementary Fig. 6) for PeV-A1 VP1 (Supplementary Fig. 7a and Supplementary Movie 2) and PeV-A3 VP3 (Supplementary Fig. 7b and Supplementary Movie 3), where the VP1 and VP3 antibodies co-localized with cells positive for dsRNA staining.

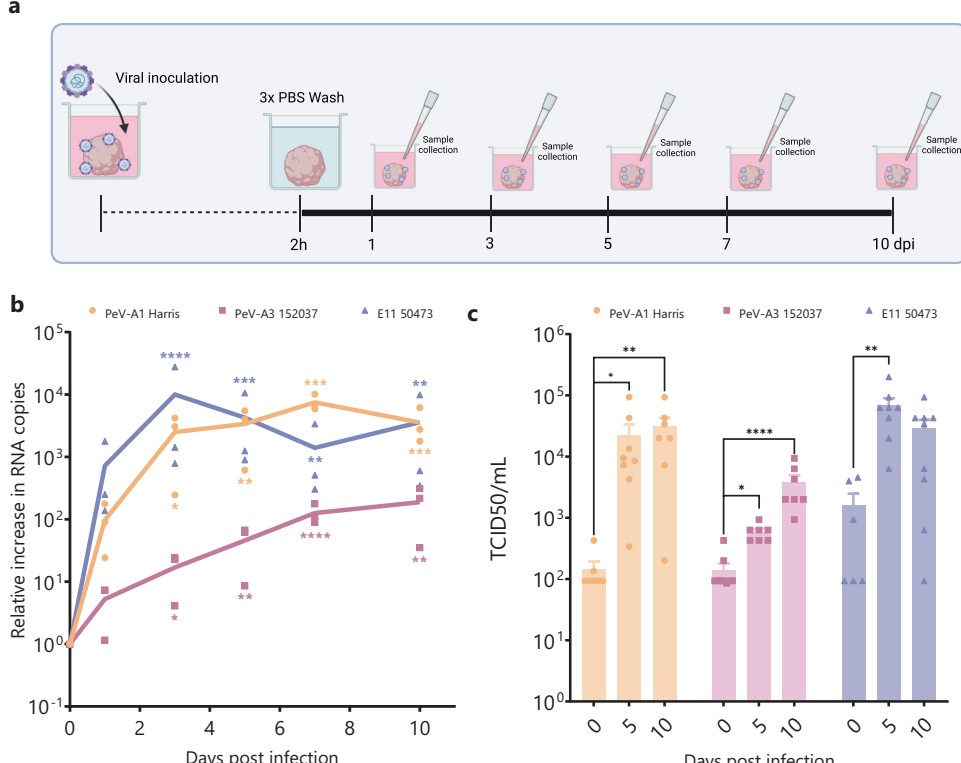

**Fig. 1 | Replication kinetics of lab-adapted strains of PeV-A1 Harris, PeV-A3 152037, and E11 50473 in unguided neural organoids. a** Schematic representation and timeline of the infection of UNOs, **b** relative increase in viral RNA copies in the supernatant at different time points, and **c** viral infectious particles from supernatant at 0, 5, and 10 dpi (days post-injection). In all cases, data correspond to the geometric mean ± geometric standard deviation (SD) of three technical replicates (individual organoids) for three batches (independent experiments) of organoids. Statistical significance was analyzed per virus using a Kruskal–Wallis test with multiple comparisons, *P value < 0.05; **P value < 0.01; ***P value < 0.001; ****P value < 0.0001. Source data are provided as a Source Data file. Biorender was used to generate (**a**).

## Lab-adapted strains of PeV-A3 and E11 induce an upregulation of inflammatory responses in comparison to PeV-A1

We previously described that PeV-A3 infection of human airway epithelium upregulated the expression of several immune-related genes at the transcriptional level such as interferon (IFN) and NF-κB signaling compared to PeV-A1[22]. Moreover, clinical data from PeV-A3 infected patients showed elevated levels of inflammatory cytokines e.g. IFN-α2, C-X-C motif chemokine ligand 10 (CXCL10), and monocyte chemoattractant protein 1 (MCP-1) in plasma[44]. However, no comparative data are available for CNS inflammation for PeV-A1 and PeV-A3.

To understand the effect of PeV-A infection on innate inflammatory responses of UNOs, we analyzed the expression of a panel of cytokines associated with PeV-A infection and key cytokines in the CNS inflammatory response[45] using RT-qPCR. PeV-A3 152037 infected organoids showed a significantly higher expression of *CXCL10* and *IFN-B1* at 5 dpi compared to PeV-A1 Harris infected organoids. This increase was maintained at 10 dpi (Fig. 3a and Supplementary Fig. 8a, b) although not significantly different. Moreover, to further look into the relation between PeV-A3 infection and the corresponding cytokine response, we measured the protein concentration of specific cytokines in the supernatant using a Luminex 10-plex assay. We found significantly higher concentrations of several inflammatory cytokines such as IFN-λ1, IFN-β, and CXCL10 in the supernatant of PeV-A3 infected UNOs both at 5 and 10 dpi as compared to MOCK infected UNOs while none of these cytokines were significantly increased upon PeV-A1 infection (Fig. 3b and Supplementary Fig. 8c, d). The upregulation pattern observed for PeV-A3 was similar to that of E11 50473 suggesting an important role for the host inflammatory response upon infection with these viruses that are associated with CNS disease.

To confirm our findings and elucidate underlying inflammatory mechanisms, we performed liquid chromatography with tandem mass spectrometry (LC-MS/MS)-based quantitative proteomic analysis following infection in UNOs at 10 dpi. The differential expression analysis identified 304, 12, and 113 differentially abundant proteins (DAPs) (adjusted P value < 0.1) upon infection with the lab-adapted strains of PeV-A1, PeV-A3, and E11, respectively, compared to the MOCK-infected samples (Fig. 3c). Of these proteins, six DAPs (ISG15, IFIT2, OAS3, MX1, IFIT3, and EDF1) were unique between PeV-A1 and PeV-A3 infection, and two DAPs (LASP1, CCDC504) were unique among PeV-A1, PeV-A3, and E11. For the full list of DAPs see Supplementary Data 1.

Gene-set enrichment analysis was further performed to identify pathways altered upon PeV-A infection and demonstrated a significant upregulation (adjusted P value < 0.1, Piano) of IFN-α and IFN-γ response in PeV-A1 Harris, PeV-A3 152037, and E11 50437 infection (Fig. 3d, e). Additionally, inflammatory response and complement pathways were only upregulated in PeV-A3 and E11-infected UNOs. Furthermore, we observed that PeV-A could significantly alter the host cell metabolism. Pathways associated with glycolysis, heme metabolism, and hypoxia were downregulated upon PeV-A1 infection. For both PeV-A1 and PeV-A3, oxidative phosphorylation was upregulated to meet the increased energy demands to activate the host antiviral response as a common feature for PeV-A infections.

Since IFN-response-associated pathways were upregulated, we looked further into the expression landscape of IFN stimulatory proteins in PeV-A infection. A previously published gene set of IFN stimulatory proteins[46] was used for the analysis, consisting of IFN-α, IFN-γ, and antiviral mechanism-related genes. We observed that 15 IFN-related proteins were significantly up- or downregulated upon either

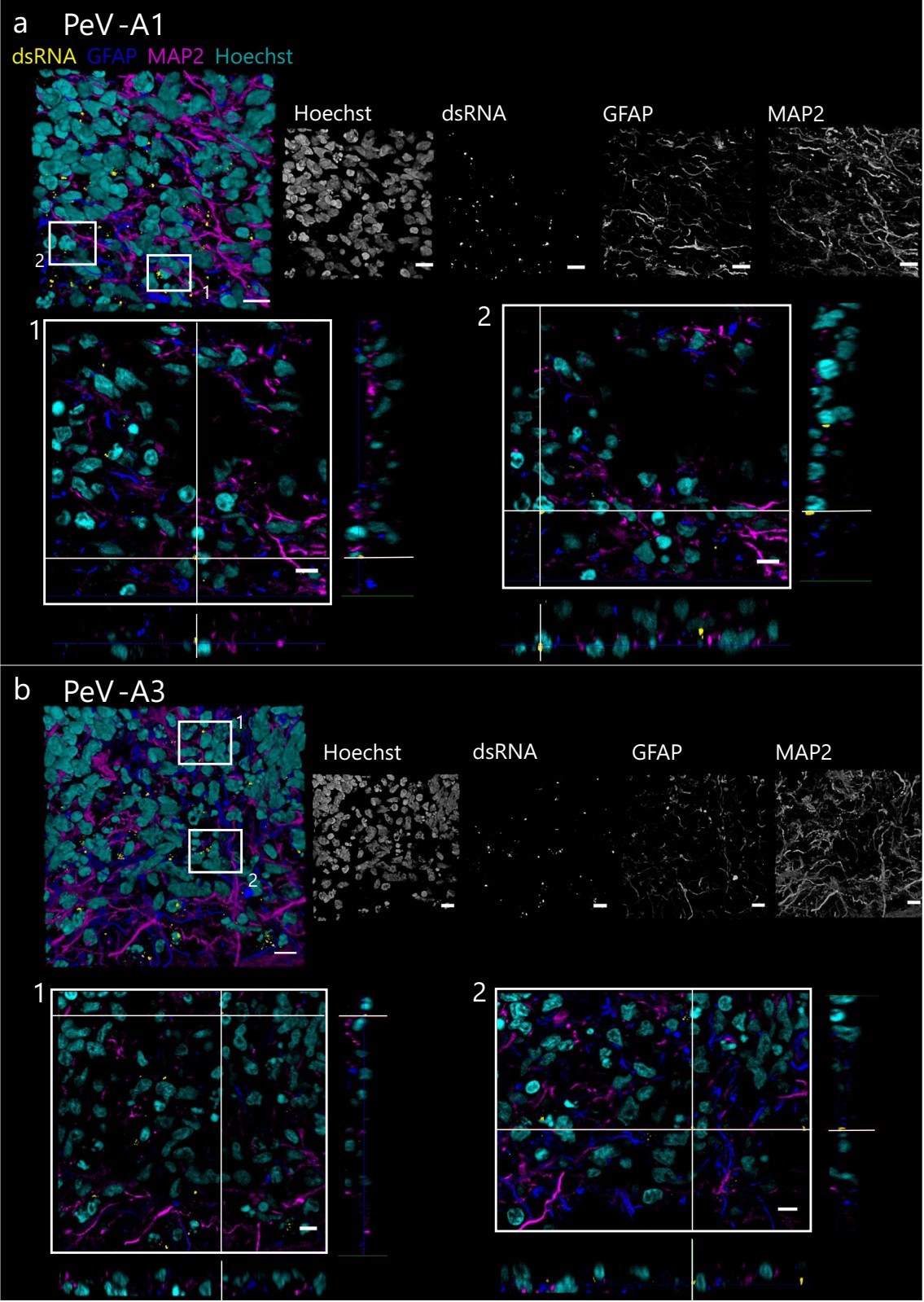

**Fig. 2 | Lab-adapted strains of PeV-A1 Harris and PeV-A3 152037 co-localize with astrocyte and neuronal markers.** Confocal Z-stacks of **a** PeV-A1 and **b** PeV-A3 infected UNOs. Labeled for nuclei (cyan) and immunolabelled for dsRNA (yellow), astrocytes (GFAP, blue), and neurons (MAP2, magenta). An orthogonal view of areas in white boxes is shown below the 3D reconstructed Z-stack. Scale bars 10 μm.

PeV-A1, PeV-A3, or E11 infection compared to MOCK (Fig. 3f and Supplementary Fig. 9a). These 15 IFN-related proteins included five IFN-stimulated genes (ISGs), namely IFIT2, IFIT3, OAS3, ISG15, and MX1[47], that were all significantly upregulated for both PeV-A1 and PeV-A3, although PeV-A3 did this to a higher degree (Fig. 3f). Further IFN and antiviral related proteins were upregulated upon infection with PeV-A3 (EIF2AK2, STAT1), or PeV-A1 (NCAM1, POM121), and downregulated upon PeV-A1 infection (EIF4G1, UBE2N, RANBP2, FLNA) (Fig. 3f). For

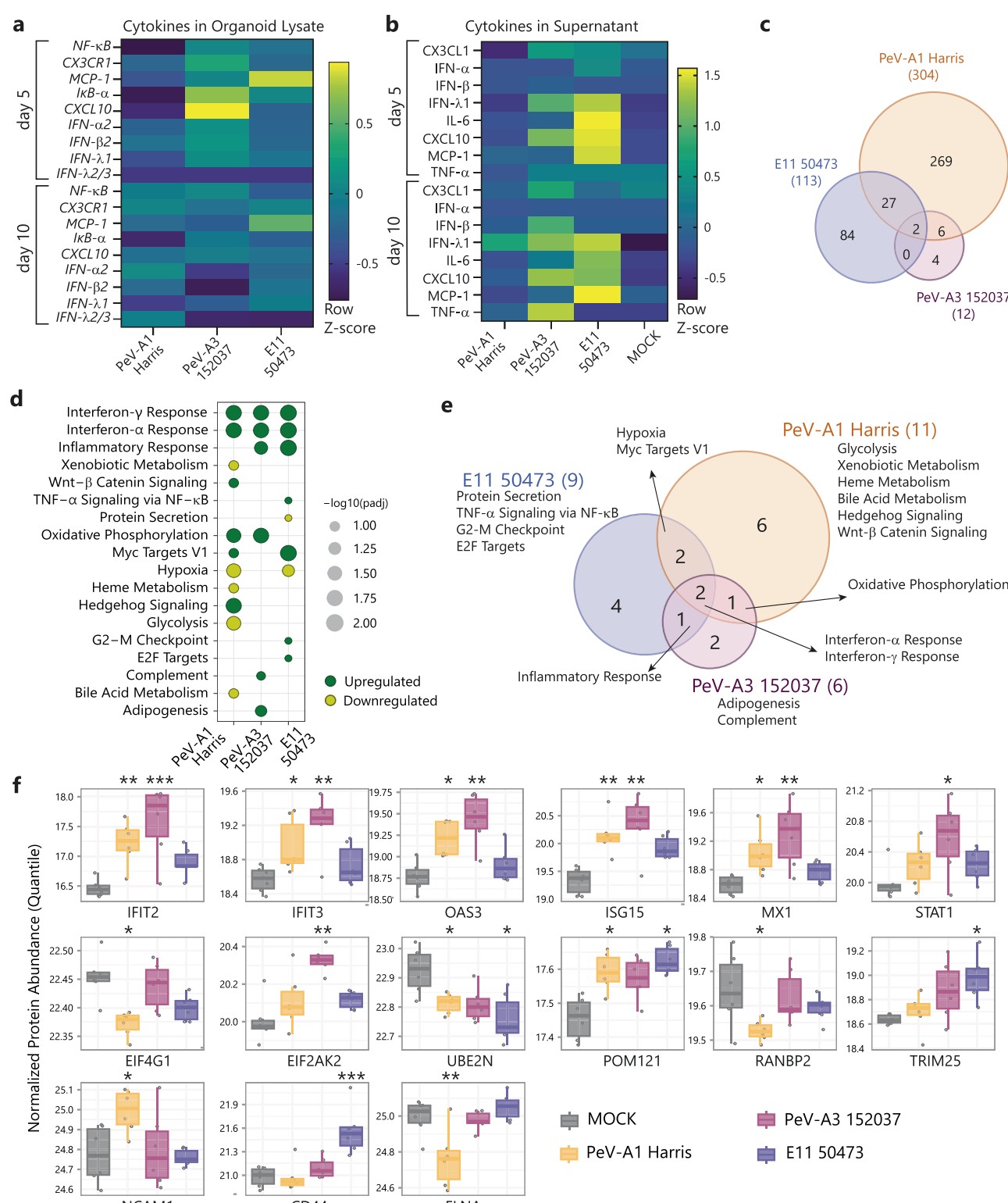

E11, no ISGs were found to be differentially expressed but other antiviral IFN-related proteins were differentially expressed (CD44, UBE2N, POM121, TRIM25).

**Blocking of the IFN pathway enhances the replication of lab-adapted PeV-A3**

As pathway analysis showed that IFN-responses were significantly upregulated upon PeV-A infection, and some of the highly upregulated cytokines, IFN-β, IFN-λ1, and CXCL10, were related to the IFN signaling,

we further characterized the role of the IFN-pathway on PeV-A infection. This was done by blocking the Janus kinase-signal transduced and activator of transcription (JAK-STAT) pathway, which activates upon IFN binding resulting in the expression of ISGs[48] using the JAK1/2 inhibitor Ruxolitinib (Rux; INCB018424)[49]. First, it was confirmed that Rux could block the JAK/STAT pathway in UNOs. Upon stimulation of organoids with 500 ng of IFN-β or IFN-λ3, Rux-treated UNOs showed downregulation of ISG expression compared to non-treated controls (Supplementary Fig. 10). Next, we determined the effect of Rux

**Fig. 3 | PeV-A3 152037 leads to upregulation of the inflammatory response. a** Heatmap representing relative gene expression of cytokines in UNOs infected with lab-adapted strains PeV-A1 Harris, PeV-A3 152037, or E11 50473 compared to MOCK infected organoids at 5 and 10 dpi by RT-qPCR. **b** Heatmap representing cytokine concentration detected in the supernatant of UNOs infected with PeV-A1 Harris, PeV-A3 152037, E11 50473, or MOCK infected at 5 and 10 dpi. **c** Venn diagram representing DAPs upon PeV-A1 Harris, PeV-A3 152037, or E11 50473 infection in organoids compared to mock infection samples (adjusted *P* value < 0.1, Limma). **d**, **e** Pathway enrichment analysis results upon PeV-A1 Harris, PeV-A3 152037, and E11 50473 infection in organoids. **d** Bubble plot represents −log10 scaled adjusted *P* value of enrichment of corresponding upregulated (green) or downregulated (yellow) pathways upon infection (adjusted **P* value < 0.1, Piano). **e** Venn diagram

representing the number of overlapping differentially expressed pathways upon infection (adjusted *P* value < 0.1, Piano). **f** Significantly altered IFN/antiviral related proteins due to PeV-A1, PeV-A3, or E11 infection compared to MOCK presented in boxplots (*adjusted *P* value < 0.1; **adjusted *P* value < 0.01; ***adjusted *P* value < 0.001, Limma). Quantile normalized protein abundance values are shown in the *Y* axis. The boxes represent the inter-quartile range and the whiskers represent minimum and maximum values. **a**, **b** All data correspond to the geometric mean ± geometric SD presented as row *Z*-score of **a** three or **b** two technical replicates (individual organoids) in three batches (independent experiments) of organoids. **c–f** Data correspond to two technical replicates (individual organoids) in three batches (independent experiments) of organoids. Source data are provided as a Source Data file.

treatment on viral ISG induction and its effect on viral replication (Fig. 4a, b). Rux treatment resulted in a downregulation of ISGs at 5 dpi for PeV-A3 (Fig. 4c), which was maintained until 10 dpi (Supplementary Fig. 11a).

At 5 dpi, we observed the effect of Rux on PeV-A3 infected organoids reflected in an increase in viral RNA copies (Fig. 4d) and infectious particles (Fig. 4e). Although the effect of blocking the JAK-STAT pathway was apparent at 10 dpi, we did not observe any significant increase in viral replication for any of the PeV-A strains at this time point (Supplementary Fig. 11b, c). Together these results indicate that IFN plays an important role in controlling PeV-A3 replication. No significant difference was found in either ISG expression or viral replication upon Rux treatment in E11 infected UNOs at day 5 (Fig. 4), but an increase in viral RNA copies was observed at day 10 (Supplementary Fig. 11b).

### Clinical isolates of PeV-A1 and PeV-A3 also infect UNOs but only PeV-A3 initiates a firm inflammatory response

As a next step, we used PeV-A clinical isolates (<5 passages) as they will be genetically closer to circulating strains[50]. This is in line with our previous report that only PeV-A3 clinical isolates could infect and replicate in the intestinal epithelium[51]. To this end, we infected UNOs 67-days-old with two clinical isolates of PeV-A1 (52967 and 51067) and PeV-A3 (178608 and 51903). To ensure reproducibility, we included the previously used lab-adapted strains for PeV-A1 (Harris), PeV-A3 (152037), and E11 (50473). The replication kinetics showed that clinical isolates of PeV-A1 and PeV-A3 could infect and replicate in UNOs to similar levels as those of lab-adapted strains (Fig. 5a). Replication of infectious virus was confirmed by an increase in TCID50 over time (Fig. 5b).

To evaluate if the clinical isolates initiated a similar inflammatory response as the lab-adapted strains, we studied DAPs at 10 dpi using quantitative proteomics. The data in Fig. 5c–f shows DAPs (nominal *P* value < 0.05, Limma) upon infection with the clinical isolates of PeV-A1 (52967 and 51067) and PeV-A3 (178608 and 51903) as well as including E11 (50473) (adjusted *P* value < 0.1, Limma). Due to high heterogeneity among UNOs infected with clinical isolates, DAP analysis after multiple hypothesis correction did not provide significantly abundant proteins. Hence, the nominal *P* value was used to infer significance[50]. The DAP analysis showed an increased number of DAPs upon infection with clinical isolates of PeV-A1 (410) compared to clinical isolates of PeV-A3 (275) (Fig. 5c, *P* value < 0.05, Limma). The results also showed a higher number of alterations in the proteomes of UNOs following infection with clinical strains of PeV-A than lab-adapted strains (Figs. 5c and 3e). For the full list of DAPs see Supplementary Data 2.

Pathway analysis showed that infection with clinical isolates of PeV-A1 and PeV-A3 resulted in the alteration of 6 and 13 pathways, respectively (Fig. 5e, adjusted *P* value < 0.1). A significant change in IFN α and γ response was observed upon infection with clinical isolates of PeV-A1, PeV-A3, and E11. However, upregulation of the inflammatory response was only observed upon PeV-A3 and E11 infection in UNOs, which correlates to the infection by lab-adapted strains. Infection with

PeV-A3 clinical isolates resulted in upregulating TNF-α signaling via the NF-κB pathway which was not observed with the lab-adapted PeV-A3 strain (Fig. 5d, e). Focusing on the IFN-regulated genes using the previously described gene set on IFN-related antiviral mechanisms, we observed 13 IFN-related proteins up- or downregulated upon infection with the PeV-A1 or PeV-A3 clinical strains (Fig. 5f and Supplementary Fig. 9b). Similar to the lab-adapted strains, we observed six ISG-related DAPs (IFIT1, IFIT2, IFIT3, OAS3, ISG15, and MX1) significantly upregulated compared to MOCK infected UNOs, upon infection with clinical isolates of PeV-A3, but not PeV-A1 (Fig. 5f). Further IFN-related DAPs upon infection with clinical isolates of PeV-A1 (EIF4E, TRIM2, ABCE1, SAMHD1) and PeV-A3 (TRIM2, MAPK3, TPR, NUP88) are shown in Fig. 5f. No overlap with E11 and the PeV-A clinical isolates was observed in IFN-related DAPs (Fig. 5f), but common pathways were upregulated for both PeV-A3 and E11 (Fig. 5d, e).

## Discussion
Despite the high prevalence of PeV-A, the difference between the most prevalent genotypes in causing CNS pathology is unknown. In this paper, we expanded the current understanding of PeV-A infection pertaining to CNS disease in humans using human brain organoids. Our data indicate that genotype-specific differences are not due to the infectivity of CNS cells as both PeV-A1 and PeV-A3 productively infected UNOs. Furthermore, the data suggest that both PeV-A3 and E11 generate a higher innate inflammatory response than PeV-A1. As strong immune responses in the CNS are associated with meningitis, encephalitis, and meningoencephalitis[51], the more profound inflammatory responses following PeV-A3 infection might explain associated CNS disease.

Previous research from our laboratory showed that in a neuroblastoma cell line, PeV-A3 was more infectious than PeV-A1[18]. However, in UNOs, we observed that PeV-A1 is more infectious compared to PeV-A3. Furthermore, our work with human-based in vitro models of the primary replication sites has shown that the cell tropism of PeV-A1 and PeV-A3 was similar in human airway epithelium but different in human intestinal epithelium[22,52]. We, therefore, hypothesized that differences in CNS disease between genotypes could be due to a differential cell tropism. Contrary to our hypothesis, we observed co-localization of dsRNA for both genotypes (as well as for E11) with the same cell types (neurons and astrocytes). Interestingly, while PeV-A1 is not considered a CNS pathogen, one outbreak of CNS symptoms caused by PeV-A1 has been reported[53]. This indicates that PeV-A1 may be able to bypass barriers to enter the CNS (such as crossing the blood–brain barrier) and other factors may play a role in PeV-A-induced CNS disease. Nevertheless, further studies on the potential of the two PeV-A genotypes to cross the BBB are warranted.

The main difference between genotypes observed in this study that could explain differential CNS disease was related to the innate inflammatory response elicited upon infection. Notably, despite the fact that PeV-A1 replicated faster and to higher titers in UNOs as compared to PeV-A3, UNOs infected with PeV-A3 showed an enhanced upregulation and production of several cytokines including IFN-λ1, CXCL10, and MCP-1. These cytokines were also upregulated in E11

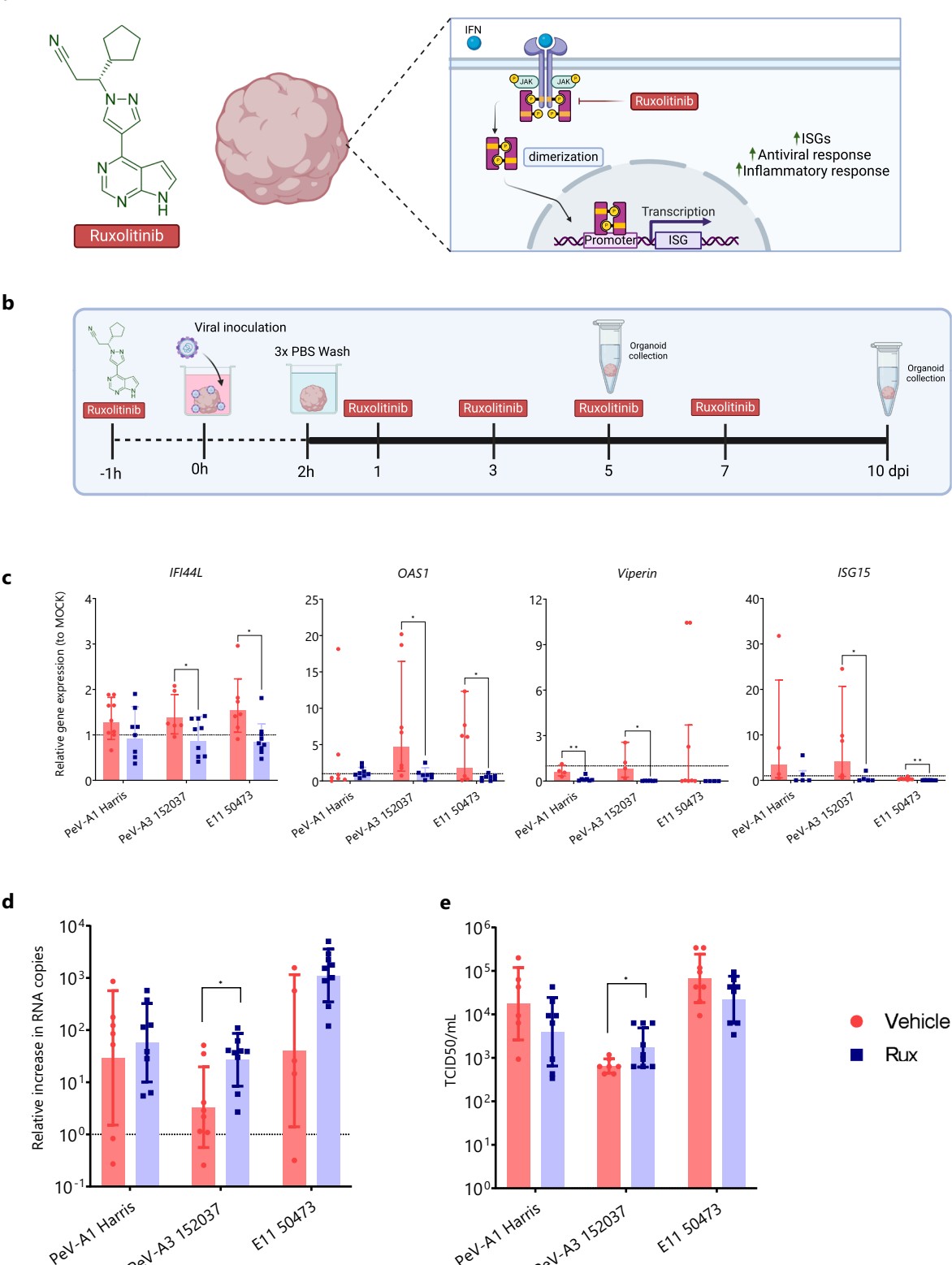

**Fig. 4 | Ruxolitinib (Rux) treatment inhibits ISG activation and enhances PeV-A3 replication. a** Schematic representation of the effect of Rux on the JAK/STAT pathway. **b** Timeline of Rux treatment on infected organoids. **c** ISGs gene expression at 5 dpi was normalized to reference genes and relative expression to MOCK-infected organoids was calculated. **d** Relative increase in RNA copies at 5 dpi for Rux or DMSO-treated organoids from supernatant samples. **e** Virus titers at 5 dpi from supernatant-collected samples of Rux or vehicle-treated organoids. Titers were determined by TCID50. In all cases, data correspond to the geometric mean ± geometric SD of three technical replicates (individual organoids) in three batches (independent experiments) of organoids. Statistical significance was determined using an unpaired two-tailed *t*-test, *$P$ value < 0.05, **$P$ value < 0.01. For **c,** values above the dashed line represent an upregulation of the gene expression relative to the MOCK. Source data are provided as a Source Data file. Biorender was used to generate (**a**, **b**).

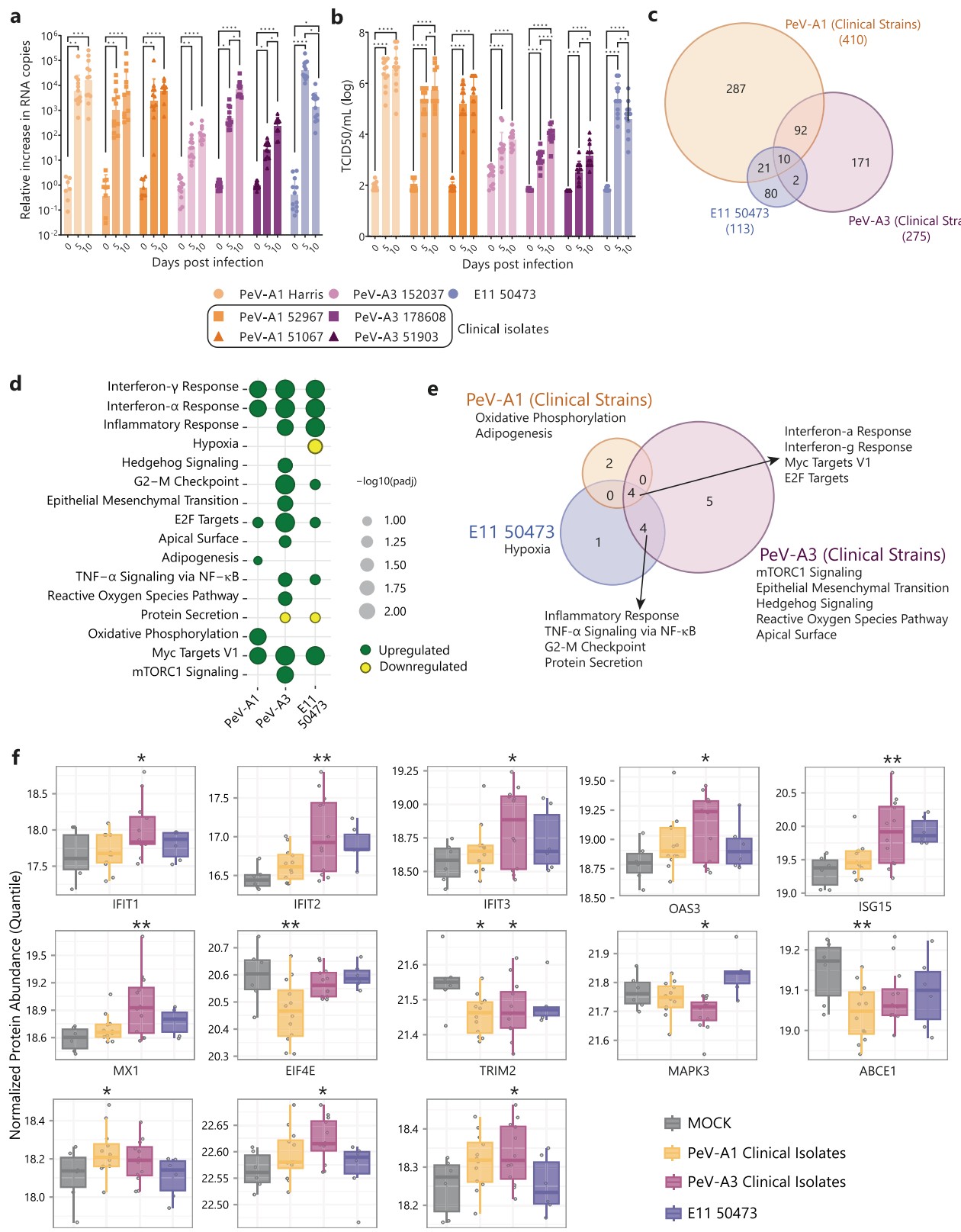

infection and have also been associated with E11 human CNS disease[54] as well as in a mouse model[55]. In line with our qPCR and Luminex data, proteomics data on several PeV-A1 and A3 strains also confirmed these findings. Although PeV-A1 infection results in a considerably larger number of DAPs than PeV-A3 infection, only PeV-A3 (and E11) infection was associated with an inflammatory response. Some of the upregulated pathways, such as the IFN-α response are also common for other viruses, such as ZIKV[43]. The higher amount of DAPs upon PeV-A1 infection could be related to its higher replication kinetics hijacking more of the cellular machinery for replication. This is supported by the enrichment of multiple pathways associated with cellular metabolic activity such as oxidative phosphorylation. However, further studies

**Fig. 5 | Clinical isolates of PeV-A1 (52967 and 51067) and PeV-A3 (178608 and 51903) showed infection and replication in UNOs and PeV-A3 initiated a succeeding inflammatory response. a** Relative increase in viral RNA copies in the supernatant at 0, 5, and 10 dpi normalized to 0 dpi. **b** Infectious viral particles in supernatant at 0, 5, and 10 dpi. **c** Venn diagram representing DAPs upon infection in organoids with clinical isolates of PeV-A1 and PeV-A3 or E11 compared to mock infection samples. **d** Bubble plot represents −log10 scaled adjusted *P* value (adjusted *P* value < 0.1, Piano) enrichment of corresponding upregulated (green) or downregulated (yellow) pathways upon infection. **e** Venn diagram representing the number of overlapping pathways up or downregulated upon infection (adjusted *P* value < 0.1, Piano*)*. **f** Significantly altered IFN-related proteins due to infection with clinical isolates of PeV-A compared to MOCK, presented in boxplots, *\*P* value < 0.05; *\*\*P* value < 0.01 obtained in Limma. Quantile normalized protein abundance values are shown in the *Y* axis. The boxes represent the interquartile range and the whiskers represent minimum and maximum values. **a, b** In all cases, data correspond to the geometric mean ± geometric standard deviation (SD) of four technical replicates of three batches of UNOs. Statistical significance was analyzed per virus using a Kruskal–Wallis test with multiple comparisons, *\*P* value < 0.05; *\*\*P* value < 0.01; *\*\*\*P* value < 0.001; *\*\*\*\*P* value < 0.0001. **c–f** Data correspond to two technical replicates (individual organoids) in three batches (independent experiments) of organoids. DAPs are differentially abundant proteins. Source data are provided as a Source Data file.

are needed to further elucidate the role of these pathways in PeV-A infection.

In terms of IFN responses, an upregulation was seen for all infections in the UNOs. However, PeV-A3 elicited a higher upregulation of ISG-related DAPs, as compared to PeV-A1. This data also suggest that, while PeV-A1 exhibited higher infectivity, PeV-A3 displayed a greater inflammatory response. The important role of the IFN pathway in controlling PeV-A3 infection is further confirmed as, upon inhibition of the JAK/STAT pathway with Rux treatment, PeV-A3 replication was enhanced. Negative correlations between the antagonism of IFN and the virulence of RNA viruses have been described previously in literature[56]. Immune modulation via the JAK/STAT pathway has previously been shown to be promising in the prevention of immune-mediated disease by SARS-CoV-2[48,57]. Understanding the molecular mechanisms behind the elicited immune response of the host to viruses such as PeV-As is essential to develop potential treatment strategies.

Studying viral behavior in vitro comes with the limitation of strains becoming lab-adapted through mutations acquired during culturing[58]. Both lab-adapted strains and clinical isolates were included in this study. It should be taken into consideration that the use of lab-adapted strains and clinical isolates can give different insights, although results were similar in this study[52,59]. While more complex than 2D models, UNOs still lack key cell types such as microglia. The contribution of this brain immune cell will be highly interesting as they are vital in neuroinflammation. Furthermore, as previously mentioned, our model lacks vasculature and the BBB (and other CNS barriers) which may be important factors in the genotype-specific CNS disease. For this study, UNOs, rather than region-specific neural organoids, were used to ensure that a broad range of CNS regions were represented. However, the inconsistency in cell number and composition that are associated with UNOs[60] can compromise reproducibility. Future studies could use guided regionalized neural organoids that are more consistent in size and composition, such as dorsal forebrain organoids[61,62].

Mechanistically, viral adaptations can avoid the activation of inflammatory pathways and production of cytokines, chemokines, and prolonged IFN-responses[63]. In the case of PeV-A1, it is possible that the virus circumvents the trigger to induce an inflammatory response, as described for many other viruses (Japanese Encephalitis virus, Rubella virus, Hendra virus, Dengue virus) that have developed ways to escape host immune responses[64]. Interference with positive feedback loops might explain why ISGs were upregulated to a lower extent after PeV-A1 than after PeV-A3 and E11 infection. Further studies on the role of PeV-A1 proteins in attenuating innate immune pathways or the inability of PeV-A3 proteins to attenuate these pathways should be performed to confirm this possibility. On a broad note, the observed inflammatory responses following PeV-A3 infection in our study align with clinical observations that include inflammatory responses e.g. meningitis, encephalitis, and meningoencephalitis[51]. Collectively, our findings align with clinical observations and suggest a role for inflammatory-mediated neurology, rather than viral replication, in PeV-A3 (and E11)

infection The mechanism behind the fact that neonates are the main human target for PeV-A3 CNS disease is still to be elucidated.

## Methods

### Cell lines and virus strains
Human colorectal adenocarcinoma cells (HT-29, ATTC HTB-38), rhesus monkey kidney cells (LLCMK2, provided by the Municipal Health Services, the Netherlands), African green monkey kidney cells (Vero, provided by the National Institute of Public Health and the Environment, RIVM, the Netherlands), and human rhabdomyosarcoma (RD, ATCC CC136) were used for virus culture. All cell lines were maintained in Eagle's minimum essential medium (EMEM, Lonza) supplemented with 8% (v/v) heat-inactivated fetal bovine serum (FBS, Sigma-Aldrich), 100 U/mL penicillin/streptomycin (Pen-Strep, Lonza), 1% (v/v) non-essential amino acids (100x, ScienceCell Research Laboratories), and 0.1% (v/v) L-glutamine (Lonza). Cell lines were incubated at 37 °C, 5% $CO_2$, and 95% humidity and passaged every 7 days using trypsin.

The following lab-adapted strains were used: PeV-A1 Harris strain was obtained from the RIVM and cultured on HT-29 cells. The PeV-A3 152037 strain[22], a Dutch isolate from 2001 adapted to cell culture, was cultured on LLCMK2 cells. The E11 50473 strain, a Dutch isolate from fecal material was cultured on Vero cells. The following clinical isolates were passaged for <10 passages in cell lines: PeV-A1 52967 and PeV-A1 51067[52] were cultured on HT-29. PeV-A3 178608 was isolated during the 2013 outbreak in Australia[65] and was cultured in LLCMK2 cells. PeV-A3 51903[52] was cultured on Vero cells. The enterovirus A71 (EV-A71) strain CA-91-480-Q () was obtained from the RIVM and cultured on RD cells.

Heat-inactivated (HI) controls were generated by incubating the virus stock in a water bath at 65 °C for 20 min and infection was performed as described previously[66].

### Human induced pluripotent stem cell culture
Human induced pluripotent stem cells (hiPSCs) (IMR90-4/WISCi004-B, WiCell) were cultured on human laminin 521 (Biolamina)-coated culture treated six-well plates and maintained in mTeSR+ medium (STEMCELL Technologies) supplemented with 1% (v/v) Pen-Strep. Cells were maintained at 37 °C with 5% $CO_2$, passaged weekly with ReLeSR™, and subcultured in mTeSR+ medium with 10 μM Y-27632 Rho Kinase (ROCK) inhibitor (Cayman Chemical Company). Lines were kept in culture with the removal of differentiated patches when necessary and regular testing for mycoplasma was performed. The maintenance and subsequent experiments with hiPSCs during maintenance and experiments were performed in accordance with relevant guidelines and regulations.

### Generation of unguided neural organoids
UNOs were generated from the IMR90 hiPSC (WiCell®) using the Cerebral Organoid Generation and Maturation kit from STEMCELL™ Technologies, which is based on the protocol for UNO generation described by Lancaster et al.[26]. In short, hiPSCs were detached into a

single cell suspension using Gentle Cell Dissociation Reagent (STEMCELL™ Technologies) and seeded in an ultra-low attachment round bottom 96-well plate (Corning) with embryoid body (EB) Formation Medium to obtain EBs. Hereafter, induction of neuroectoderm was obtained using Induction Medium (STEMCELL™ Technologies) followed by expansion of neuroepithelia by embedding EBs in ESC-qualified Matrigel (Corning) and culturing in Expansion Medium (STEMCELL™ Technologies). On day 10 the organoids were placed on an orbital shaker (66 rpm) in Maturation Medium (STEMCELL™ Technologies) and the medium was refreshed every 3–4 days until infection at day 67.

### Infection of unguided neural organoids

UNOs from three independent batches were infected in technical triplicates or quadruplicates with $10^5$ TCID50 per mL of the different viruses. Individual organoids were placed on a round bottom 96-well plate coated with anti-adherence rinsing solution (STEMCELL™ Technologies) and 100 μL of the virus inoculum were added. Organoids were incubated for 2 h at 37 °C with 5% $CO_2$, washed three times with phosphate buffer saline (PBS, Lonza), and moved to a freshly coated 48-well plate with 500 μL of Maturation Medium (STEMCELL™ Technologies). After 10 min incubation, the 0 h time-point was collected, and the medium was replenished. Collection with full medium replenishment was repeated at 1, 3, 5, 7, and 10 dpi where samples from the same organoid were taken. Back titrations of viral inoculums were also performed to confirm comparable inoculation titers using TCID50.

### RT-qPCR

RNA was isolated from 25 μL of the collected supernatant using the Bioline Isolate II RNA mini kit (Meridian Bioscience®) following the manufacturer´s instructions. Equal volumes of the eluted RNA were used for reverse-transcription and 5 μL of the cDNA was used for reverse-transcription quantitative PCR (RT-qPCR). qPCR was performed on a CFX Connect Real-Time PCR Detection System (Bio-Rad) using software CFX Maestro 1.1, and $C_q$ values were transformed into viral genome copies using a standard curve with known concentrations of the viral genomes. For RT-qPCR primers see Supplementary Table 1.

To analyze cytokine expression UNOs were harvested in RLY lysis buffer (Bioline Isolate II RNA mini kit (Meridian Bioscience®)) and stored at −80 °C until RNA isolation. The sample was thoroughly homogenized by vortexing and resuspended by pipetting before RNA was isolated. The same protocol as described previously was used for RNA extraction, cDNA synthesis, and RT-qPCR. Cytokine upregulation was measured using primer sets (see Supplementary Table 1, Biolegio) where expression of the target gene was normalized to reference genes. The combination of RPLP0 and RPLP2 was chosen as the most stably expressed set of reference genes in both MOCK and virus-infected organoids using Normfinder[67] (NormFinder Excel Add-In MS Excel 2003 v0.953). Gene expression was normalized using the method[68] using the geometric mean of both reference genes. Infected samples were normalized to uninfected control to visualize the effect of infection on cytokine expression in the UNOs. Z-scores were calculated by subtracting the mean of the condition (in rows) from each value, followed by dividing the result by the standard deviation (SD) of that population.

### TCID50

Supernatant samples (25 μL) of multiple time points were titrated for each virus, where PeV-A1 strains were titrated on HT-29, PeV-A3 152037, and 178608 were titrated on LLCMK2 and E11 50473, and PeV-A3 51903 were titrated on Vero cells. Briefly, ten-fold dilutions of each sample were performed and seeded in a 96-well plate (50 μL), the appropriate cells were added (200 μL) and incubated for 10 days until readout. For the readout, the cells were examined for the appearance

of cytopathic effect, and the TCID50 was calculated using the Reed and Muench method[69] and normalized to the 0 h time-point to determine the increase of infectious particles over time.

### Immunofluorescence staining

Organoids or cell lines were fixed at 5 and 10 dpi with 4% (v/v) formaldehyde (Sigma-Aldrich) in PBS for 30 min at room temperature (RT). After fixation, organoids were washed three times with PBS and incubated in 30% (w/v) sucrose (Merck) by overnight incubation at 4 °C. The organoids were embedded in optimal cutting temperature compound (OCT, Tissue Tek) snap frozen on dry ice, and stored at −80 °C until sectioning. Twenty-micrometer sections were cut using a cryostat (NX71, Thermo Fisher Scientific) and collected on SuperFrost Plus slides (Thermo Fisher Scientific). Sections were stored at −80 °C until staining. For immunostaining, sections were blocked for 2 h at RT in a blocking solution consisting of 10% (v/v) SeaBlock Blocking Buffer (Thermo Fisher Scientific) with 1% (v/v) Triton X-100 (Sigma) in PBS. After blocking, primary antibodies (Supplementary Table 2) were added in 1:1 blocking solution:PBS and incubated overnight at 4 °C. Sections were washed three times with PBS for 5 min, and incubated with secondary antibody (Supplementary Table 2) solution and Hoechst (Thermo Fisher Scientific) at RT for 1 h. Samples were quenched using ReadyProbes Tissue Autofluorescence Quenching kit (Invitrogen, kit) and incubated for 5 min, followed by three PBS washes. Finally, slides were mounted with glass coverslips using Pro-Long Gold Antifade Mounting Medium (Invitrogen). UNOs were imaged using a Leica TCS SP8-X microscope and Leica LAS AF Software (Leica Microsystems) or EVOS M5000 microscope (Thermo Fisher Scientific). Z-stacks were also taken, and 3D reconstructions were made using the LAS-X 3D software (Leica Microsystems) and ImageJ 1.50I.

### LC-MS/MS-based quantitative proteomics

The organoid specimens were thawed on ice and resuspended in 50 μL of lysis buffer, comprising 40 μL of 8 M urea in Tris-HCl and 10 μL of 0.5% ProteaseMax (Promega). Subsequently, 1 μL of 100× protease inhibitor cocktail (Roche) was introduced, and the samples underwent sonication in a water bath for 10 min. Following sonication, 49 μL of Tris-HCl, pH 8.5 was added into each sample to make a final volume of 100 μL. The samples were then sonicated with a probe for 40 s at 20% amplitude and on/off pulse 2/2 s. The lysate was centrifuged at 4 °C for 10 min and 16,000×$g$. The resulting supernatant was carefully transferred to LoBind Eppendorf tubes.

For the protein estimation, each sample was diluted at a 1:5 ratio, and protein estimation was performed using Pierce™ BCA protein assay kit (ThermoFisher Scientific) following the manufacturer's protocol. The volume corresponding to 25 μg of protein was transferred to a new tube and supplemented with lysis solution to achieve a final volume of 25 μL. This 25 μL of protein was further supplemented with 7.5 μL of acetonitrile (ACN) and 42.5 μL of Tris-HCl pH 8.5, bringing the final volume to 75 μL. The protein underwent reduction with 1.5 μL of 0.5 M dithiothreitol in Tris-HCl buffer, followed by incubation at 37 °C for 1 h. Subsequently, alkylation was performed with 4.5 μL of 0.5 M iodoacetamide at RT in the dark for 30 min. The alkylation reaction was terminated by adding 3 μL of 0.5 M dithiothreitol in Tris-HCl buffer. Next, 1 μg of sequencing-grade modified trypsin (Promega) was added to the samples and allowed to incubate for 16 h at 37 °C. The digestion process was halted by adding 4.5 μL of concentrated formic acid (FA) and incubating the solutions at RT for 5 min. The samples were then purified on a C18 Hypersep plate with a 40 μL bed volume (Thermo Fisher Scientific) and subsequently dried using a vacuum concentrator.

The TMTpro 18-plex (Thermo Fisher Scientific) was used to label the samples and multiplexed. The combined TMT-labeled biological replicates were fractionated by high-pH reversed-phase after

 

dissolving in 50 µL of 20 mM ammonium hydroxide and were loaded onto an Acquity bridged ethyl hybrid C18 UPLC column (2.1 mm inner diameter × 150 mm, 1.7 µm particle size, Waters), and profiled with a linear gradient of 5–60% 20 mM ammonium hydroxide in ACN (pH 9.0) over 48 min, at a flow rate of 200 µL/min. The chromatographic performance was monitored with a UV detector (Ultimate 3000 UPLC, Thermo Fisher Scientific) at 214 nm. Fractions were collected at 30 s intervals into a 96-well plate and combined in 12 samples concatenating 8–8 fractions representing peak peptide elution.

The peptide fractions in solvent A (0.1% FA in 2% ACN) were analyzed by LC-MS/MS as described before[70], except that mass spectra were acquired on a Q Exactive HF hybrid quadrupole Orbitrap mass spectrometer (Thermo Fisher Scientific) ranging from $m/z$ 375 to 1700 at a resolution of $R = 120,000$ (at $m/z$ 200) targeting $1 \times 10^6$ ions for maximum injection time of 80 ms, followed by data-dependent higher-energy collisional dissociation (HCD) fragmentations of precursor ions with a charge state 2+ to 7+, using 45 s dynamic exclusion. The tandem mass spectra of the top 18 precursor ions were acquired with a resolution of $R = 60,000$, targeting $2 \times 10^5$ ions for a maximum injection time of 54 ms, setting quadrupole isolation width to 1.4 Th and normalized collision energy to 34%. The sample identities with the corresponding TMT channels can be found in Supplementary Table 3.

Acquired raw data files were analyzed using Proteome Discoverer v3.0 (Thermo Fisher Scientific) with Sequest HT search engine against the *Homo sapiens* consensus protein database (UniProt TaxID=9606 v2023-02-09). A maximum of two missed cleavage sites were allowed for full tryptic digestion while setting the precursor and the fragment ion mass tolerance to 10 ppm and 0.02 Da, respectively. Carbamidomethylation of cysteine was specified as a fixed modification. Oxidation on methionine, deamidation of asparagine and glutamine, as well as acetylation of N-termini and TMTpro were set as dynamic modifications. Initial search results were filtered with 1% FDR using the Percolator node in Proteome Discoverer. Quantification was based on the reporter ion intensities normalizing on the total peptide amount in each channel in Proteome Discoverer.

### Bioinformatics analysis
Due to multiple batches of TMT runs, we checked the normality distribution and identified that the distribution of the samples needed additional normalization methods. Further, the data was normalized using different methods implemented in the *R* package NormalyzerDE v1.12.0[71] to improve data distribution characteristics. Data normalized using the quantile method outperformed other methods, thus used for all downstream analyzes Missing values were imputed using *R* package impute v1.68.0[72] sing the $k$-nearest neighbors method ($k = 10$, rowmax = 0.8, colmax = 0.8, and maxp = 1500). Batch effect correction was performed using the *R* function ComBat from the package sva v3.42.0. The data was analyzed for differential abundance analysis using the *R* package Linear Models for Microarray Data (Limma) v3.50.0[73]. The relative viral load of the samples was added to the Limma model matrix while comparing virus strains to account for possible bias. Pathway analysis was performed using the Hallmark Gene sets from the human Molecular Signatures Database (MSigDB)[74]. Collections obtained from enrichr libraries[75] and *R* package Piano v2.10.0[76] (nPerm = 1000, geneSetStat = mean, and signifMethod = geneSampling). The input file for Piano included the *P* value and log2 fold change value of all the proteins following the differential protein abundance (DAP). Piano provides the gene set results into directionality classes of the resulting pathways based on the gene pattern. The multiple hypothesis test correction in Limma and Piano was performed using the Benjamini–Hochberg (BH) method. The abundance level of IFN-associated protein was investigated using custom curated gene sets, namely antiviral mechanism by ISGs, IFN-γ signaling, and IFN α/β signaling with 205 IFN-regulated genes[46].

### Ruxolitinib treatment
UNOs were pretreated with 5 µM or vehicle dimethyl sulfoxide (DMSO, Santa Cruz Biotechnology) and incubated for 1 h before infection at 37 °C. After pre-treatment, organoids were stimulated with 500 ng IFN β (R&D Systems), or IFN-λ3 (R&D Systems), or infected as described previously with PeV-A1, PeV-A3, or E11. Treatment with 5 µM Rux/vehicle was continued throughout the 10 days post-infection with every medium change at 1, 3, 5, 7, and 10 dpi.

### ProcartaPlex Multiplex Immunoassay
To detect cytokines present in supernatant samples of (un)infected brain organoids, a customized 10-plex Luminex® assay was used (ProcartaPlex Multiplex Immunoassay, Invitrogen, Bio-Plex Manager Software v6.2). Samples were lysed with 12.5% (v/v) cell lysis buffer (Invitrogen) to inactivate viruses and the measurement was performed following the manufacturer's instructions. Fluorescence was measured using a Luminex (R&D) and from this cytokine concentrations were calculated using the provided standard curve in the kit. Values that were below the lower limit of detection (LLOD) were replaced by the $LLOD/\sqrt{2}$[77]. The data was presented as $Z$-scores of the raw data for each row.

### Data visualization and statistical analysis
All statistical analysis other than LC-MS/MS-based quantitative proteomic analysis was performed using GraphPad Prism 8 (GraphPad Software Inc.). Experiments were performed in three independent organoid batches in triplicates (unless otherwise indicated). Data are presented as geometric mean ± geometric SD. The specific statistical tests performed for each analysis are indicated in the corresponding figure legend. Differences were considered significant when the *P* value was <0.05.

To visualize overlapping protein expression between conditions Venn Diagrams were produced using the online tool Interactivenn[78] and Adobe Illustrator 2023. R package ggplot2 v3.4.2 was used to generate boxplots and bubble plots while heatmaps were made using Graphpad or R package ComplexHeatmap v2.10.0[79].

### Reporting summary
Further information on research design is available in the Nature Portfolio Reporting Summary linked to this article.

## Data availability
The raw data supporting the conclusions of this article is available in Figshare (doi: 10.21942/uva.21982610, 10.21942/uva.25244491, and 10.21942/uva.25244521). Sequencing results are available on GenBank (accession numbers BankIt: E11 OR886062 and PeV-A OR886056-OR886061). The mass spectrometry proteomics data have been deposited to the ProteomeXchange Consortium via the PRIDE partner repository with the dataset identifier PXD047238. Source data are provided with this paper.

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

## Acknowledgements

All schematic representations were created with BioRender.com. We acknowledge Daisy Picavet and Dr. Ron Hoebe from the core facility Cellular Imaging of the Amsterdam UMC (Amsterdam, the Netherlands) for their help with confocal imaging. We also thank the Municipal Health Services and the National Institute of Public Health, and the Environment (RIVM) for the supply of the cell lines, the molecular laboratory staff for typing all of the samples, and Dr. Kimberley S.M. Benschop and Dr. Jeroen Cremer of the RIVM for sequence analysis. We acknowledge the support from Dr. Akos Vegvary, Proteomics Biomedium, and the Core for Systems Infection Biology, Karolinska Institute. Finally, we acknowledge AII Work visit grant for facilitating a work visit between the Amsterdam UMC and Karolinska Institutet. This research was funded by the European Union's Horizon 2020 Research and Innovation Programme under the Marie Sklowdowska-Curie Grant Agreement OrganoVIR (grant 812673, KW/DP) and GUT-VIBRATIONS (grant 953201, KW/DP), the PPP Allowance (Focus-on-Virus, DP/KW) made available by Health Holland, Top Sector Life Sciences & Health, to the Amsterdam UMC, location Amsterdam Medical Center to stimulate public–private partnerships, and the Van Herk group through a donation to Amsterdam UMC, location Academic Medical Center (DP). UN acknowledges support from the Swedish Research Council grants 2021-00993 (UN) and 2021-01756 (UN).

## Author contributions

Conceptualization: P.C., I.G.-R., D.P., A.S., and K.W. Data curation: P.C., I.G.-R., A.T.A., L.M., E.F., J.D., G.K., V.S., and R.V.-S. Formal analysis: P.C., I.G.-R., A.T.A. Funding acquisition: A.S., D.P., and K.W. Investigation: P.C., I.G.-R., A.T.A., L.M., E.F., J.D., G.K., V.S., and R.V.-S. Methodology: P.C., I.G.-R., A.T.A., L.M., E.F., J.D., G.K., and R.V.-S. Project administration: A.S., D.P., and K.W. Supervision: C.C., R.V.-S., U.N., D.P., A.S., and K.W. Validation: P.C., I.G.-R. Visualization: P.C., I.G.-R. Writing of the original draft: P.C., I.G.-R., and A.S. Writing, review, and editing: P.C., I.G.-R., A.T.A., L.M., E.F., J.D., G.K., C.C., V.S., R.S.-V., U.N., D.P., A.S., and K.W. As described by CRediT-Contributor

Roles Taxonomy (casrai.org). All authors contributed to the article and approved the submitted version.

## Competing interests

J.A.D. and R.S. are employees and/or shareholders of UniQure Bio-pharma B.V. The remaining authors declare that the research was conducted in the absence of any commercial or financial relationships that could be construed as a potential conflict of interest.
