## [Peer Review File · Nature Communications]

Parechovirus infection in human brain organoids: host innate inflammatory response and not neuro-infectivity correlates to neurologic diseaseEditorial Note: This manuscript has been previously reviewed at another journal that is not operating a transparent peer review scheme. This document only contains reviewer comments and rebuttal letters for versions considered at Nature Communications.

Reviewers' Comments:

Reviewer #1:

Remarks to the Author:

All my major concerns have been adequately addressed. The authors performed experiments with clinical strains and confirmed their observations, consolidating the findings in three strains of each parechovirus. Quantitative proteomics supports previous findings and highlights the importance of inflammatory responses as a major driver of neurological diseases.

The only remaining concern is the absence of a quantitative proteomic dataset. The list of targets in Supplementary Text 1 and 2 does not reflect the quantitative results and adjusted p-values.

Reviewer #3:

Remarks to the Author:

Dear authors,

This is an interesting paper with difficult cell (tissue) model. Unfortunately, I found many mistakes in the writing, which I hope can be corrected. I also urge you to describe clearly how the virus levels were balanced before infection, and how outcome was measured. I am puzzled that dsRNA only was used in the primary analyses while virus protein-specific antibodies were used in Supplementary images. The functionality of the antibodies requires explanation. No parechovirus antibodies in the market, not done by you? I am not fully confident if the data allow the claims but comparative studies are always challenging. Further comments below:

In my version I have Dutch in many sentences, there is also different capitalization of letters as well as different styles in the list of references. There is even one reference in duplicate. Please, check Abstract and Introduction

Picornaviruses in general are written in lower case letters, such as parechovirus A3 (as genotype).

Lines 23 and 45

Line 37: Parechoviruses are NOT known as Parechovirus A, since the latter is taxonomical term.

Line 44: echoviruses (generic name for these viruses)

Line 44: genus Enterovirus (in italics since this is taxonomical term)

Line 72: Parechoviruses are not known to infect small animals. Could you elaborate this finding.

Line 78: Could you elaborate the UNOs and immune system. Are there immune cells in the interior? How well this model simulates brain tissue? Might be good to mention here and not in the end as a limitation.

Line 88: Can you mention whether there are any studies using PeV-infected brain tissues to support your findings similarly to CMV?

Results

Line 119: something in Dutch? There are many places where similar sentence is shown. Typo?

Figure 1 does not indicate the cell line used to propagate the infected virus. More importantly, if this is about tropism, one cannot use cells to compare since tropism (differential receptor use) certainly will affect the outcome. How was the virus amount set between the types before infection?

Figure 2. dsRNA stain only, while VP1-/VP3-antibodies were used in the Supplement. Could you explain this apparent discrepancy?

Line 164: I previously called upon brain samples from PeV-A3 infected patients. Was this only from PeV-A3 and are there any related data for PeV-A1?

Lines 211-216 seem like Discussion

The results end with the claim that PeV-A3 and echovirus 11 upregulate common signaling pathways. It might be useful to make a comment on the most relevant ones in respect of CNS disease symptoms in respect to other viruses for which there are more information.

Line 344: Are you aware of studies where virus has been isolated in CNS tissue (besides CSF samples) in the brain? Thus, is BBB blocking virus entry but not inflammatory responses or are those derived from virus-infected cells?

Line 354: Based on the definition, echoviruses do not infect mice, so how come echovirus 11 has been shown to be associated with CNS disease in mouse model (ref. 56). Could you explain the findings in the paper to elaborate the relationship with cytokine upregulation, CNS disease echovirus 11 and parechovirus 3.

Line 360 this sentence is somewhat undermining your findings. If mentioned, what more should be done, and more importantly why it was not? No known relation between viral functions (proteins?) and immune responses?

References

-are not in the same format

-are the references #32 and #40 the same?

Reviewer #4:

Remarks to the Author:

I was asked to referee the protein Mass spec portion of this manuscript. The protein mass spec experiment describes the relative quantitation of proteins in unguided neural organoids (UNOs) infected with two isolates of Parechovirus: A1 and A3 (non-neurovirulent/neurovirulent, respectively). A third virus (echovirus 11) was included also.

I have two overall impressions of the study: (a) the authors description of their work is inadequate for assessment. This was compounded by the extensive use of custom scripts/workflows. (b) The authors possibly had difficulty drawing clear conclusions from the protein mass spec data: In this regard, it appears the authors have relied upon protein abundance changes of just a few percent, and jump in an non-obvious fashion between P-values of 0.05 and 0.1, between "Adjusted P-value" and "nominal p-value", and between day-5 and day-10 timepoints. At least it seemed this way.

The manuscript was essentially devoid of any description of mass spec methodology, referring instead to reference 70 for all methods, which are quite extensive. To render the manuscript clearly comprehensible the following are required:

(1) Authors should state explicitly that the following were done precisely as described in reference 70: Protein solubilization, protein concentration determination (whose method was not described in ref 70), Amount of sample digested, Alkylation and trypsin digestion conditions, peptide desalting conditions, TMT labeling, model of nanoLC and spectrometer used, nanoLC column dimensions and packing material, nanoLC flow rate and gradient, mass spec instrument method, database and associated search parameters, method for determining TMT reporter ion intensities.

(2) Importantly, to enable the reader to deduce the design of protein mass spec experiment(s) even in the absence of a clear description, authors should provide a supplementary table of sample

identities vs. all TMT channels employed.

(3) The authors seem to describe a TMT experiment with an LFQ workflow. For example, they describe "normalization" as a central step in data processing, but for a multiplexed approach such as TMT no normalization step is required intrinsically (with LFQ it is, because each sample is subjected to a separate nanoLC-MS/MS run). Are they normalizing each reporter channel individually to total protein for that sample? Same issue applies for the imputation of missing values: In a multiplexed experiment, undetectable signal for a given reporter channel can be regarded as zero intensity (abundance) because all channels are fragmented simultaneously. Actual missing quant values only arise where separate nanoLC-MS/MS runs are combined, since a different set of parental ions may have been picked for tandem MS in each run. Imputation is not even described in ref. 70. Were the authors simply given the wrong method description in error? Or are the authors normalizing and imputing because they are merging multiple TMT18plex runs against one another? It was unclear because the scope of TMT channels (above) was not provided.

(4) Replicates: Legend to Fig. 3 c-f says: "Data represents [sic] two technical replicates in three batches of organoids.". "Technical replicates" refers to what? Same vial injected twice to the spectrometer? Duplicate tryptic digestions of the same sample? Duplicate samples taken from same organoid preparation? Was the whole TMT experiment duplicated or did technical replicates have duplicate channels within the same TMT expt?

(5) Statistics: Methods include a section "Data visualization and statistical analysis". This section starts "All statistical analysis other than proteomic analysis was performed ..." etc. Does this section relate to protein MS? Because it also states: "Experiments were performed in three independent organoid batches in triplicates" (which may refer to MS also, this is unclear, though # of samples is key to understanding the statistics). In addition, it states "Differences were considered significant when the P-value was <0.05 ". However, descriptions of protein mass spec (lines 185, 207, 282, 309 and Fig. 3c/d/e legend, Fig. 5 d/e legend, Supplementary Text 1 and 2) variously use P-value < 0.1 or mix 0.1 and the more standard 0.05. In some cases, it is not clear whether P-values are being used to indicate confidence in protein ID or confidence in quantitation (differential protein abundance): Emphasizing this point, no Volcano plots are shown. Anyway, FDR-based scoring would seem superior to P-values for assessing quant differences.

(6) Figures are not great: Figs. 3f/5f Y axes are labeled "Normalized expression". Normalized to what? Fig. 3f/5f legend explains "compared to mock". But it is not so, because mock is shown as a separate (gray) bar in each of the graphs of the Fig., with "Normalized expression" values for the mock ranging from ~ 16.5 to 25. Should we assume the maximum fold change (e.g. IFIT2) w.r.t. mock was therefore $18/16.5 = 1.09$ fold? Although apparently statistically significant, is there any precedent for a 9% increase in protein abundance having significant biological effect? The standard in the protein mass spec field is typically 2-fold. In Fig. 3f box & whiskers plots presumably represent mean/SD/range?

In Fig. 3a,b the color bar scale is not annotated. Is this fold-change? Relative to mock? Log2 fold-change? Log10?

In Fig. 3d, for the given scale (" $-\log_{10}(\text{pad})$ "), the legend states " $-\log_{10}$ scaled adjusted P-value of enrichment". Presumably this represents the probability that quant values for the subset of proteins with the given functional annotations in Fig. 3d were drawn randomly from the distribution of quant values for all proteins? If so, then 1.25 (the smallest dot on the scale bar) represents $p = 0.056$. But some dots in Fig. 3d appear smaller (less significant) than the accompanying scale, representing, perhaps, $p = 0.3$?

(7) Miscellaneous missing methods: Line 283: "Multiple hypothesis correction" is not described in M&M. Supp. Fig. 8 shows a hierarchical clustering, whose method seems not described anywhere, along with a color bar "LDC" which is not described anywhere.

Other (minor) points:

Line 163 should say "at the transcriptional level".

What are referred to throughout as DEPs or "differentially expressed proteins" are strictly speaking differentially abundant proteins, since differences could be at the level of degradation.

Line 188 (semantic point): "six DEPs (ISG15, IFIT2, OAS3, MX1, IFIT3, and EDF1) were common between PeV-A1 and PeV-A3 infection, and two DEPs (LASP1, CCDC504) were in common between PeV-A1, PeV-A3 and E11. ". The latter two (common to all three) would also be common to A1 and A3.

Line 266: Here: "We have previously reported that only PeV-A3 clinical isolates were able to infect and replicate in the intestinal epithelium compared to lab-adapted strains".... "Unlike in the intestinal epithelium, the lab-adapted PeV-A3 strains were able to replicate in UNOs." ... "we infected UNOs of 67 days old with two clinical isolates of PeV-A1 (52967 and 51067) and PeV-A3 (178608 and 51903)". The flow of logic seems somewhat tortured: Intestinal epithelium infection with lab-adapted strain A3 seems to provide no obvious rationale for, or connection to, the proposed experiment to study UNOs infected with clinical isolates of strain A1, unless all combinations of host and virus were included together as internal controls in the new protein MS experiment.

Line 169: This sentence seems self-contradictory: "PeV-A3 152037 infected organoids showed a significantly higher expression of CXCL10 and IFN-B1 at 5 dpi compared to PeV-A1 Harris infected organoids that was maintained at 10 dpi (Figure 3a and Supplementary Figure 7a-b) although not significantly different."

A point-by-point response to the reviewers comments is detailed below. Our response is provided in blue to differentiate from the reviewers comments. The line numbers refer to the clean version (without track-changes).

REVIEWER COMMENTS

Reviewer #1 (Remarks to the Author)

All my major concerns have been adequately addressed. The authors performed experiments with clinical strains and confirmed their observations, consolidating the findings in three strains of each parechovirus. Quantitative proteomics supports previous findings and highlights the importance of inflammatory responses as a major driver of neurological diseases.

We thank the reviewer for the positive assessment of our manuscript.

Comment 1: The only remaining concern is the absence of a quantitative proteomic dataset.

Response: There was a delay in the uploading of the proteomic dataset when the manuscript was initially sent out for review but these datasets can now be found on the PRIDE partner repository using the dataset identifier "**PXD047238**".

Comment 2: The list of targets in Supplementary Text 1 and 2 does not reflect the quantitative results and adjusted p-values.

Response: We identified that the descriptions of these supplementary texts were not clear. These lists of differentially abundant proteins (DAPs), previously referred to as differentially expressed proteins (DEPs), pertain to the Venn diagrams in the manuscript. The descriptions have now been adjusted.

Reviewer #3 (Remarks to the Author)

Dear authors,

This is an interesting paper with difficult cell (tissue) model. Unfortunately, I found many mistakes in the writing, which I hope can be corrected.

We appreciate the detailed feedback provided by the reviewer and we have now adjusted the mistakes in writing that have been raised.

Comment 1: I also urge you to describe clearly how the virus levels were balanced before infection, and how outcome was measured.

Response: Input virus titres were $1E5$ TCID₅₀/mL, calculated based on the titre of the viral stock. The diluted inoculum, as well as the stock, in each experiment was back titrated to confirm the titres. In all instances, the back titre was within the expected range of $1E5$ TCID₅₀/mL. This is now described in the methods in lines 451-453. In all cases, infection outcome was measured by RT-qPCR and TCID₅₀, in addition to other readouts such as immunofluorescence or quantitative proteomics.

Comment 2: I am puzzled that dsRNA only was used in the primary analyses while virus protein-specific antibodies were used in Supplementary images. The functionality of the antibodies requires explanation. No parechovirus antibodies in the market, not done by you? I am not fully confident if the data allow the claims but comparative studies are always challenging.

Response: In the previous version of our manuscript, double stranded RNA (dsRNA) antibody was used for virus detection in the immunofluorescence images. In our experience, dsRNA antibodies are a standardly used in immunofluorescence imaging of RNA viruses with a double stranded RNA phase in its replication cycle¹⁻³. The PeV-A specific antibodies were added as an additional control to validate the use of dsRNA, specifically in the detection of PeV-A. Although the VP1 and VP3 antibodies targeting PeV-A are referred to as custom antibodies, they are manufactured by Thermo Fisher Scientific. They are custom only to the extent of us providing the peptide sequence information against which these antibodies were raised. For all intents and purposes, these antibodies are akin to any commercial antibody that can be purchased from the manufacturer. Post-purchase validation of antibody specificity was performed by us and has been included as *Supplementary Figure 6*. Moreover, the peptide sequence against which these antibodies were raised has been added to *Supplementary Table 2* and should enable access for comparative studies. Nevertheless, we believe that the dsRNA antibodies will be an easier route for comparative studies and for this reason, they have been retained in our primary analysis.

Comment 3: In my version I have Dutch in many sentences, there is also different capitalization of letters as well as different styles in the list of references. There is even one reference in duplicate. Please, check

Response: The Dutch text arise due to formatting errors on upload and have been corrected. Duplicated reference has been removed.

Abstract and Introduction: Picornaviruses in general are written in lower case letters, such as parechovirus A3 (as genotype). Lines 23 and 45.

Response: Genotypes are now appropriately written with a lower case.

Comment 4: Line 37: Parechoviruses are NOT known as Parechovirus A, since the latter is taxonomical term.

Response: This sentence has been adjusted to reflect its intended meaning.

Comment 4: Line 44: echoviruses (generic name for these viruses) and Line 44: genus Enterovirus (in italics since this is taxonomical term)

Response: This has now been adjusted accordingly.

Comment 5: Line 72: Parechoviruses are not known to infect small animals. Could you elaborate this finding.

Response: They cannot naturally infect small animals through the oral-fecal route. In this case, the work by Jan et al., 2021 demonstrates PeV-A infection in the brain of neonatal mice through intracranial inoculation⁴.

Comment 6: Line 78: Could you elaborate the UNOs and immune system. Are there immune cells in the interior? How well this model simulates brain tissue? Might be good to mention here and not in the end as a limitation.

Response: In the UNO protocol, microglia can sometimes be found⁵ but we do not expect them at the organoid age used in our study⁶. We have now added a sentence (lines 83-84) to clarify this.

Comment 7: Line 88: Can you mention whether there are any studies using PeV-infected brain tissues to support your findings similarly to CMV?

Response: To our knowledge there are no patient-derived specimens that have been reported for PeV-A.

Comment 8: Results. Line 119: something in Dutch? There are many places where similar sentence is shown. Typo?

Response: The cross-references were not saved properly when uploading the final document. They have now been adjusted indicating the correct label of the Figure across the manuscript.

Comment 9: Figure 1 does not indicate the cell line used to propagate the infected virus. More importantly, if this is about tropism, one cannot use cells to compare since tropism (differential receptor use) certainly will affect the outcome. How was the virus amount set between the types before infection?

Response: The specific cell lines in which the viruses were propagated are described in the Materials and Methods section. It has been recently reported that PeV-A genotypes A1 to A6 use MYADM as an entry receptor⁷ in different cell lines. We agree that this does not preclude adaptation in our cell lines but the use of low-passage clinical isolates, with minimal changes, helps limit the effect of culture adaptation. Virus input was based on the TCID50 measured in the corresponding lines. While these can vary based on the cell line used, the input TCID50 between genotypes can also be vastly different in the organoids. Given the lack of established methodologies for measuring viral titres in the organoids, we have used the best practice possible in minimizing variations in viral input.

Comment 10: Figure 2. dsRNA stain only, while VP1-/VP3-antibodies were used in the Supplement. Could you explain this apparent discrepancy?

Response: Please see our response to comment 2.

Comment 11: Line 164: I previously called upon brain samples from PeV-A3 infected patients. Was this only from PeV-A3 and are there any related data for PeV-A1?

Response: Best available is plasma and cerebrospinal fluid data from PeV-A3 infected patients. We are not aware of similar reports for PeV-A1.

Comment 12: Lines 211-216 seem like Discussion

Response: Line 215 has now been removed.

Comment 13: The results end with the claim that PeV-A3 and echovirus 11 upregulate common signaling pathways. It might be useful to make a comment on the most relevant ones in respect of CNS disease symptoms in respect to other viruses for which there are more information.

Response: We have added a sentence regarding this in lines 352-353.

Comment 14: Line 344: Are you aware of studies where virus has been isolated in CNS tissue (besides CSF samples) in the brain? Thus, is BBB blocking virus entry but not inflammatory responses or are those derived from virus-infected cells?

Response: To the best of our knowledge this has not been reported.

Comment 15: Line 354: Based on the definition, echoviruses do not infect mice, so how come echovirus 11 has been shown to be associated with CNS disease in mouse model (ref. 56). Could you explain the findings in the paper to elaborate the relationship with cytokine upregulation, CNS disease echovirus 11 and parechovirus 3.

Response: The authors in reference 56 used a transgenic mouse model that lacks type I IFN response and has been modified to express the human neonatal Fc receptor (the receptor for echoviruses)⁸. E11 is able to infect the brain of these mice. In this study they also find a differential abundance of cytokines in the brain of the infected mice that is similar to some of the cytokines found to be differentially abundant in our study.

Comment 16: Line 360 this sentence is somewhat undermining your findings. If mentioned, what more should be done, and more importantly why it was not? No known relation between viral functions (proteins?) and immune responses?

Response: We do not believe that highlighting limitations undermines our finding but rather emphasizes the context of the findings. It is our firm belief that any model system has its limitations that must be taken into account when interpreting the findings from that model.

Comment 17: References -are not in the same format

Response: This has been adjusted.

Comment 18: -are the references #32 and #40 the same?

Response: Duplicates have been removed.

Reviewer #4 (Remarks to the Author)

I was asked to referee the protein Mass spec portion of this manuscript. The protein mass spec experiment describes the relative quantitation of proteins in unguided neural organoids (UNOs) infected with two isolates of Parechovirus: A1 and A3 (non-neurovirulent/neurovirulent, respectively). A third virus (echovirus 11) was included also.

I have two overall impressions of the study:

(a) the authors description of their work is inadequate for assessment. This was compounded by the extensive use of custom scripts/workflows.

(b) The authors possibly had difficulty drawing clear conclusions from the protein mass spec data: In this regard, it appears the authors have relied upon protein abundance changes of just a few percent, and jump in an non-obvious fashion between P-values of 0.05 and 0.1, between "Adjusted P-value" and "nominal p-value", and between day-5 and day-10 timepoints. At least it seemed this way.

Response: We thank the reviewer for taking the time to go over our manuscript thoroughly and for providing insightful feedback on the MS methodology. We recognize the importance of presenting our study transparently and we have added additional information to improve our reporting.

Comment 1: The manuscript was essentially devoid of any description of mass spec methodology, referring instead to reference 70 for all methods, which are quite extensive. To render the manuscript clearly comprehensible the following are required:

(1) Authors should state explicitly that the following were done precisely as described in reference 70: Protein solubilization, protein concentration determination (whose method was not described in ref 70), Amount of sample digested, Alkylation and trypsin digestion conditions, peptide desalting conditions, TMT labeling, model of nanoLC and spectrometer used, nanoLC column dimensions and packing material, nanoLC flow rate and gradient, mass spec instrument method, database and associated search parameters, method for determining TMT reporter ion intensities.

Response: We acknowledge your concern about the (in)adequacy of the description of our work for assessment. We have extensively published on TMT-based proteomic data analysis for several viral infections⁹⁻²⁰. To avoid duplication and extensive self-citation, we referred to only one of our previous publications where the methods were described extensively. Nevertheless, to ensure that relevant information for evaluating the LC-MS and quantitative proteomics are readily available, we have expanded this in the methods with a separate section (504-578). This includes detailed analysis parameters for the individual tools used for bioinformatics along with a rationale for their use. The tools used in our study are standard in the field, open-sourced, and highly cited in the proteomics data analysis.

Comment 2: Importantly, to enable the reader to deduce the design of protein mass spec experiment(s) even in the absence of a clear description, authors should provide a supplementary table of sample identities vs. all TMT channels employed.

Response: This has now been added as Supplementary Table 3.

Comment 3: The authors seem to describe a TMT experiment with an LFQ workflow. For example, they describe "normalization" as a central step in data processing, but for a multiplexed approach such as TMT no normalization step is required intrinsically (with LFQ it is, because each sample is subjected to a separate nanoLC-MS/MS run). Are they normalizing each reporter channel individually to total protein for that sample? Same issue applies for the

imputation of missing values: In a multiplexed experiment, undetectable signal for a given reporter channel can be regarded as zero intensity (abundance) because all channels are fragmented simultaneously. Actual missing quant values only arise where separate nanoLC-MS/MS runs are combined, since a different set of parental ions may have been picked for tandem MS in each run. Imputation is not even described in ref. 70. Were the authors simply given the wrong method description in error? Or are the authors normalizing and imputing because they are merging multiple TMT18plex runs against one another? It was unclear because the scope of TMT channels (above) was not provided.

Response: The MS methodology has been updated as mentioned in comment 1. However, we, respectfully, disagree with the comment that “for a multiplexed approach such as TMT no normalization step is required intrinsically”. We firmly believe that the normalization (often multiple) depends on the data and the experimental plan.

For the current multi-batch dataset, we employed several normalization steps. The first normalization is an equal amount of proteins (25 µg each) were used. The second normalization step is performed by Proteome discoverer, normalizing, based on the reporter ion intensities, the total peptide amount in each channel. The data then dictates the choice for additional normalization. In this regard, data distributions and density plots are checked. If the data distributions do not have well aligned box plots then it would indicate the need for further normalization. Despite improved sample distributions following normalization, (unexpected) batch effects may arise and additional normalization for internal reference scaling (IRS) using pooled samples may be warranted (https://pwilmart.github.io/IRS_normalization/understanding_IRS.html).

In our present dataset, we observed that the data distribution is inappropriate (see plot below). Therefore, we employed additional normalization methods that resulted in appropriate Quantile data. This quantile normalized data was used for downstream analysis.

Figure 1- Data normalization using Quantile Normalization Method

Comment 4: Replicates: Legend to Fig. 3 c-f says: “Data represents [sic] two technical replicates in three batches of organoids.”. “Technical replicates” refers to what? Same vial injected twice to the spectrometer? Duplicate tryptic digestions of the same sample? Duplicate samples taken from same organoid preparation? Was the whole TMT experiment duplicated or did technical replicates have duplicate channels within the same TMT expt?

Response: Technical replicates refer to individual organoids from each batch (= independent organoid generation and subsequent independent infection). This helps overcome any bias that may arise due to heterogeneity inherent to the unguided differentiation protocol. This has been clarified in the corresponding figure captions.

Comment 5: (5) Statistics: Methods include a section “Data visualization and statistical analysis”. This section starts “All statistical analysis other than proteomic analysis was performed ...” etc. Does this section relate to protein MS? Because it also states: “Experiments were performed in three independent organoid batches in triplicates” (which may refer to MS also, this is unclear, though # of samples is key to understanding the statistics). In addition, it states “Differences were considered significant when the p -value was <0.05 ”. However, descriptions of protein mass spec (lines 185, 207, 282, 309 and Fig. 3c/d/e legend, Fig. 5 d/e legend, Supplementary Text 1 and 2) variously use p -value < 0.1 or mix 0.1 and the more standard 0.05. In some cases, it is not clear whether p -values are being used to indicate confidence in protein ID or confidence in quantitation (differential protein abundance):

Emphasizing this point, no Volcano plots are shown. Anyway, FDR-based scoring would seem superior to p -values for assessing quant differences.

Response: The statement on statistical analysis and p -values are not related to MS nor proteomic analysis. This statement has been amended for clarity. For instance, in figure 5d and e, adjusted $p < 0.1$ was used for gene set enrichment analysis (GSEA) in Piano. The test method used is now mentioned following the adjusted p -values. Boxplots were chosen to show the differences in the individual proteins as it is more informative on sample distribution. The p -values in the boxplot refer to the confidence in differential protein abundance.

Comment 6: (6) Figures are not great: Figs. 3f/5f Y axes are labeled "Normalized expression". Normalized to what? Fig. 3f/5f legend explains "compared to mock". But it is not so, because mock is shown as a separate (gray) bar in each of the graphs of the Fig., with "Normalized expression" values for the mock ranging from ~16.5 to 25. Should we assume the maximum fold change (e.g. IFIT2) w.r.t. mock was therefore $18/16.5 = 1.09$ fold? Although apparently statistically significant, is there any precedent for a 9% increase in protein abundance having significant biological effect? The standard in the protein mass spec field is typically 2-fold. In Fig. 3f box & whiskers plots presumably represent mean/SD/range?

Response: As mentioned in comment 3, the protein abundance values were normalized using the quantile method and is now explained in the methods section. The quantile normalized values are shown on the y-axis and is now specified in the figure captions. Figure 3F and 5F show differential comparisons to the MOCK in all cases.

The fold change calculation performed by the reviewer is conceptually correct but cannot be exactly translated to the absolute changes given that this is quantile normalized data and fold changes is log fold changes (logFC). These fold change values were calculated using LIMMA algorithm after adjustment for the viral load. LIMMA fits a linear model to the log-expression values to compute log fold change (logFC). It should also be judge by the statistical significance that helps to distinguish true biological changes from random variability.

As the reviewer calculated an example, IFIT2, here is the calculation:

In mock vs PeV-A3 the logFC was 1.12 (after adjustment of the viral load) which is equivalent to 2.29 fold ratio (i.e. double) and level of significance adjusted p -value 0.0006. While in mock vs PeV-A1 the logFC was 0.76 which is equivalent to 1.81 fold ratio and level of significance adjusted p -value 0.009.

Factors such as precision weights, inter-gene correlation, missing values, and multiple model terms are considered while the linear model calculates the fold change. A detailed explanation of the algorithm can be found in the original publication of Limma R package referenced in the method section. In Figure 3F, the box represents inter-quartile range and whiskers represent minimum and maximum values.

Comment 6-1: In Fig. 3a,b the color bar scale is not annotated. Is this fold-change? Relative to mock? Log2 fold-change? Log10?

Response: Data in Figure 3 a and b corresponds to the Z-scores of the raw data for each row of the heatmaps. This means the raw data is subtracted by the mean of that row and divided by the standard deviation of that row. Further normalization or log transformations were not performed. This is now clarified in the methods and added to the figure legend. In Figure 3 a and b the color scale bar is now annotated as "Row Z-score".

Comment 6-2: In Fig. 3d, for the given scale ("-Log₁₀(pad)"), the legend states "-log₁₀ scaled adjusted P-value of enrichment". Presumably this represents the probability that quant values for the subset of proteins with the given functional annotations in Fig. 3d were drawn randomly from the distribution of quant values for all proteins? If so, then 1.25 (the smallest dot on the scale bar) represents $p = 0.056$. But some dots in Fig. 3d appear smaller (less significant) than the accompanying scale, representing, perhaps, $p = 0.3$?

Response: R package Piano was used for GSEA to identify pathways. Piano uses protein level statistics (p -value of differential protein abundance and direction in log₂ scaled fold changes as input to determine significance and directionality of enrichment. Significant is assessed using gene sampling strategy where the gene labels are randomized n PerM times (n PerM=1000) and the gene set statistics are recomputed so that background distribution for each original gene set is acquired. We used Piano to obtain three classes of gene set significance describing different aspects of regulation directionality: distinct-directional, mixed-directional, and non-directional. We considered distinct-directional p -values (distinct-directional up and distinct-directional down) of gene sets to define the significance and regulation directionality of gene sets. Distinct-directional p -values are computed from gene statistics with sign information (p -value and Log₂foldchange).

Comment 7: Miscellaneous missing methods: Line 283: "Multiple hypothesis correction" is not described in M&M. Supp. Fig. 8 shows a hierarchical clustering, whose method seems not described anywhere, along with a color bar "LDC" which is not described anywhere.

Response: Benjamini-Hochberg procedure was used and is now mentioned in the methods section. The hierarchical clustering was performed using the method "complete" from the R function hclust. The clustering distance was computed using the Euclidean method. It is now added in the figure legend. We found no colour bar titled "LDC" in the manuscript.

Other (minor) points:

Comment 8: Line 163 should say "at the transcriptional level".

Response: Sentence has been modified in the manuscript.

Comment 9: What are referred to throughout as DEPs or "differentially expressed proteins" are strictly speaking differentially abundant proteins, since differences could be at the level of degradation.

Response: We agree with the reviewer; therefore, DEP changed to differentially abundant protein (DAP) throughout the manuscript.

Comment 10: Line 188 (semantic point): "six DEPs (ISG15, IFIT2, OAS3, MX1, IFIT3, and EDF1) were common between PeV-A1 and PeV-A3 infection, and two DEPs (LASP1, CCDC504) were

in common between PeV-A1, PeV-A3 and E11. ". The latter two (common to all three) would also be common to A1 and A3.

Response: The "common" now changed to "unique" mentioned here corresponds to the intersection in set theory, which is represented commonly using Venn diagram. In set theory, $A \cap B \cap C \neq A \cap B$ where A, B, and C are independent sets of elements.

Comment 11: Line 266: Here: "We have previously reported that only PeV-A3 clinical isolates were able to infect and replicate in the intestinal epithelium compared to lab-adapted strains".... "Unlike in the intestinal epithelium, the lab-adapted PeV-A3 strains were able to replicate in UNOs." ... "we infected UNOs of 67 days old with two clinical isolates of PeV-A1 (52967 and 51067) and PeV-A3 (178608 and 51903)". The flow of logic seems somewhat tortured: Intestinal epithelium infection with lab-adapted strain A3 seems to provide no obvious rationale for, or connection to, the proposed experiment to study UNOs infected with clinical isolates of strain A1, unless all combinations of host and virus were included together as internal controls in the new protein MS experiment.

Response: It is correct that the rationale for the use of clinical isolate of A3 cannot be extended to A1. Nevertheless, in addition to consistency, the use of clinical isolates of A1 is relevant as these reflect the genetic diversity and make up of circulating virus strains. Lab-adapted strains, by definition, have adapted to long term cell culture leading to loss of diversity and potential alterations that may not represent clinical phenotype. In some instances, drastically different results (as per observations in the intestinal epithelium)²¹ can be observed between lab-adapted and clinical isolate. The phrasing of this section has been adjusted so that the reasoning is not mixed up.

Comment 12: Line 169: This sentence seems self-contradictory: "PeV-A3 152037 infected organoids showed a significantly higher expression of CXCL10 and IFN- β 1 at 5 dpi compared to PeV-A1 Harris infected organoids that was maintained at 10 dpi (Figure 3a and Supplementary Figure 7a-b) although not significantly different."

Response: The sentence did not convey the intended meaning and has been corrected.

References

- 1 Moshiri, J., Craven, A. R., Mixon, S. B., Amieva, M. R. & Kirkegaard, K. Mechanosensitive extrusion of Enterovirus A71-infected cells from colonic organoids. *Nature Microbiology* **8**, 629-639 (2023). <https://doi.org/10.1038/s41564-023-01339-5>
- 2 Good, C., Wells, A. I. & Coyne, C. B. Type III interferon signaling restricts enterovirus 71 infection of goblet cells. *Sci Adv* **5**, eaau4255 (2019). <https://doi.org/10.1126/sciadv.aau4255>
- 3 Uchida, L. *et al.* The dengue virus conceals double-stranded RNA in the intracellular membrane to escape from an interferon response. *Scientific Reports* **4**, 7395 (2014). <https://doi.org/10.1038/srep07395>
- 4 Jan, M. W., Su, H. L., Chang, T. H. & Tsai, K. J. Characterization of Pathogenesis and Inflammatory Responses to Experimental Parechovirus Encephalitis. *Front Immunol* **12**, 753683 (2021). <https://doi.org/10.3389/fimmu.2021.753683>

- 5 Ormel, P. R. *et al.* Microglia innately develop within cerebral organoids. *Nat Commun* **9**, 4167 (2018). <https://doi.org:10.1038/s41467-018-06684-2>
- 6 Chiaradia, I. & Lancaster, M. A. Brain organoids for the study of human neurobiology at the interface of in vitro and in vivo. *Nature Neuroscience* **23**, 1496-1508 (2020). <https://doi.org:10.1038/s41593-020-00730-3>
- 7 Watanabe, K. *et al.* Myeloid-associated differentiation marker is an essential host factor for human parechovirus PeV-A3 entry. *Nature Communications* **14**, 1817 (2023). <https://doi.org:10.1038/s41467-023-37399-8>
- 8 Wells, A. I. & Coyne, C. B. An In Vivo Model of Echovirus-Induced Meningitis Defines the Differential Roles of Type I and Type III Interferon Signaling in Central Nervous System Infection. *J Virol* **96**, e0033022 (2022). <https://doi.org:10.1128/jvi.00330-22>
- 9 Saccon, E. *et al.* Cell-type-resolved quantitative proteomics map of interferon response against SARS-CoV-2. *iScience* **24**, 102420 (2021). <https://doi.org:10.1016/j.isci.2021.102420>
- 10 Appelberg, S. *et al.* Dysregulation in Akt/mTOR/HIF-1 signaling identified by proteo-transcriptomics of SARS-CoV-2 infected cells. *Emerg Microbes Infect* **9**, 1748-1760 (2020). <https://doi.org:10.1080/22221751.2020.1799723>
- 11 Akusjärvi, S. S. *et al.* Integrative proteo-transcriptomic and immunophenotyping signatures of HIV-1 elite control phenotype: A cross-talk between glycolysis and HIF signaling. *iScience* **25**, 103607 (2022). <https://doi.org:10.1016/j.isci.2021.103607>
- 12 Krishnan, S. *et al.* Metabolic Perturbation Associated With COVID-19 Disease Severity and SARS-CoV-2 Replication. *Mol Cell Proteomics* **20**, 100159 (2021). <https://doi.org:10.1016/j.mcpro.2021.100159>
- 13 Neogi, U. *et al.* Multi-omics insights into host-viral response and pathogenesis in Crimean-Congo hemorrhagic fever viruses for novel therapeutic target. *Elife* **11** (2022). <https://doi.org:10.7554/eLife.76071>
- 14 Ambikan, A. T. *et al.* Multi-omics personalized network analyses highlight progressive disruption of central metabolism associated with COVID-19 severity. *Cell Syst* **13**, 665-681.e664 (2022). <https://doi.org:10.1016/j.cels.2022.06.006>
- 15 Appelberg, S. *et al.* Nucleoside-Modified mRNA Vaccines Protect IFNAR(-/-) Mice against Crimean-Congo Hemorrhagic Fever Virus Infection. *J Virol* **96**, e0156821 (2022). <https://doi.org:10.1128/jvi.01568-21>
- 16 Svensson Akusjärvi, S. *et al.* Peripheral blood CD4(+)CCR6(+) compartment differentiates HIV-1 infected or seropositive elite controllers from long-term successfully treated individuals. *Commun Biol* **5**, 357 (2022). <https://doi.org:10.1038/s42003-022-03315-x>
- 17 Svensson Akusjärvi, S. *et al.* Role of myeloid cells in system-level immunometabolic dysregulation during prolonged successful HIV-1 treatment. *Aids* **37**, 1023-1033 (2023). <https://doi.org:10.1097/qad.0000000000003512>
- 18 Ambikan, A. T. *et al.* Systems-level temporal immune-metabolic profile in Crimean-Congo hemorrhagic fever virus infection. *Proc Natl Acad Sci U S A* **120**, e2304722120 (2023). <https://doi.org:10.1073/pnas.2304722120>
- 19 Mikaeloff, F. *et al.* Trans cohort metabolic reprogramming towards glutaminolysis in long-term successfully treated HIV-infection. *Commun Biol* **5**, 27 (2022). <https://doi.org:10.1038/s42003-021-02985-3>
- 20 Chen, X. *et al.* Type-I interferon signatures in SARS-CoV-2 infected Huh7 cells. *Cell Death Discov* **7**, 114 (2021). <https://doi.org:10.1038/s41420-021-00487-z>

- 21 García-Rodríguez, I., van Eijk, H., Koen, G., Pajkrt, D., Sridhar, A. & Wolthers, K. C. Parechovirus A Infection of the Intestinal Epithelium: Differences Between Genotypes A1 and A3. *Front Cell Infect Microbiol* **11**, 740662 (2021). <https://doi.org/10.3389/fcimb.2021.740662>

Reviewers' Comments:

Reviewer #3:

Remarks to the Author:

I find the responses to reviewer criticism adequate.

Reviewer #4:

Remarks to the Author:

Authors seem to have provided all the missing information. Even simply revealing this to be a triple 16-plex TMT experiment (critical information - supp. Table 3), all the normalization and imputation analysis now falls into place. The new section of M&M has some grammatical/spelling errors but the production editor should be able to find them.